# The ATPase hCINAP regulates 18S rRNA processing and is essential for embryogenesis and tumour growth

Dongmei Bai[1,2], Jinfang Zhang[1,2], Tingting Li[1,2], Runlai Hang[3,4], Yong Liu[1,2], Yonglu Tian[2], Dadu Huang[2], Linglong Qu[1,2], Xiaofeng Cao[3,4], Jiafu Ji[5] & Xiaofeng Zheng[1,2]

Dysfunctions in ribosome biogenesis cause developmental defects and increased cancer susceptibility; however, the connection between ribosome assembly and tumorigenesis remains unestablished. Here we show that hCINAP (also named AK6) is required for human 18S rRNA processing and 40S subunit assembly. Homozygous $CINAP^{-/-}$ mice show embryonic lethality. The heterozygotes are viable and show defects in 18S rRNA processing, whereas no delayed cell growth is observed. However, during rapid growth, CINAP haploinsufficiency impairs protein synthesis. Consistently, hCINAP depletion in fast-growing cancer cells inhibits ribosome assembly and abolishes tumorigenesis. These data demonstrate that hCINAP reduction is a specific rate-limiting controller during rapid growth. Notably, hCINAP is highly expressed in cancers and correlated with a worse prognosis. Genome-wide polysome profiling shows that hCINAP selectively modulates cancer-associated translatome to promote malignancy. Our results connect the role of hCINAP in ribosome assembly with tumorigenesis. Modulation of hCINAP expression may be a promising target for cancer therapy.

[1] State Key Laboratory of Protein and Plant Gene Research, School of Life Sciences, Peking University, Beijing 100871, China. [2] Department of Biochemistry and Molecular Biology, School of Life Sciences, Peking University, Yiheyuan Road No. 5, Beijing 100871, China. [3] State key Laboratory of Plant Genetics and National Center for Plant Gene Research, Institute of Genetics and Developmental Biology, Chinese Academy of Sciences, Beijing 100101, China. [4] College of Life Sciences, University of the Chinese Academy of Sciences, Beijing 100039, China. [5] Key Laboratory of Carcinogenesis and Translational Research, Department of Gastrointestinal Surgery, Peking University Caner Hospital and Institute, Beijing 100142, China. Correspondence and requests for materials should be addressed to X.Z. (email: xiaofengz@pku.edu.cn).

Ribosome biogenesis is an essential and highly orchestrated process in eukaryotic cells, which includes synthesis and processing of pre-ribosomal RNAs, coordinated ribosome protein synthesis, ribosome assembly and transport[1]. Ribosome assembly is very dynamic and closely linked to growth control[2,3]. Increased ribosomal demand, as indicated by enlarged nucleoli, has been characterized as an independent prognostic marker for malignant transformation[4]. The relationship between ribosome biogenesis and cancer development is particularly noteworthy, because alterations in ribosome synthesis have long been considered as merely a byproduct of cancer malignancy[4]. This view was challenged in recent years by studies, indicating that genetic alterations in ribosomal machinery are associated with human pathology and increased susceptibility to cancers[1,5]. Among identified genetic alterations in ribosomal machinery, mutation of RPS19 in patients with Diamond–Blackfan Anemia produces defects in 18S rRNA maturation and 40S subunit assembly[6–9]. In other cases, reducing the abundance of RPL24 limited Myc-induced lymphomagenesis[10]. Moreover, haploinsufficiency of RPS14 was identified as the cause of the $5q^-$ syndrome[11].

The correlation between ribosomal abnormalities and tumorigenesis was strengthened by the evidence that some oncogenes and tumour suppressors are involved in direct regulation of ribosome biogenesis[12,13]. The oncogene c-Myc functions as a coactivator of RNA polymerase I and III in the transcription of rRNA[14]. p53 inhibits RNA polymerase I transcription by hindering the formation of a complex necessary for the recruitment of RNA polymerase I to the rRNA gene promoter[1,5,15]. These findings raise the possibility that oncogenes and tumour suppressors may affect cancer progression partly by controlling ribosome production[16]. As ribosome biogenesis are tightly correlated with translational regulation, increased cancer susceptibility associated with altered ribosomal activity may be due to an increased protein synthesis rate and selection of specific cancer-associated messenger RNAs for translation[10,17,18], as in the case of congenital dyskeratosis[19]. The mechanisms by which ribosome biogenesis drives cancer formation is currently garnering intense interest, because protein synthesis underlies cell growth and proliferation[20]. Therefore, identification of novel factors involved in ribosome biogenesis and the exact mechanisms by which such factors regulate ribosome biogenesis and alter tumour susceptibility is crucial.

Human coilin-interacting nuclear ATPase protein hCINAP, also known as adenylate kinase 6, is highly conserved in eukaryotes. hCINAP is a typical α/β protein with a structure common to adenylate kinases[21]. Adenylate kinases play important roles in nucleotide metabolism by catalysing reversible transfer of the γ-phosphate of ATP to AMP, forming two molecules of ADP[22]. Atypically, hCINAP also contains a Walker B motif characteristic of ATPases[23]. In human cells, hCINAP participates in the formation of Cajal bodies, the nuclear particles involved in maturation of small nuclear snRNPs[24]. Recently, hCINAP was also shown to regulate p53 activity via the HDM2-p53 pathway[25,26]. Fap7, the yeast homologue of hCINAP, is essential for ribosome assembly and yeast growth[27]. In humans, the physiological function of hCINAP and the link between its enzymatic activities and biological processes crucial for tumour growth remain uncertain.

Here we show that hCINAP is required for human 18S rRNA processing. hCINAP is highly expressed in cancers and promotes cancer cell growth through selectively upregulates the translation of cancer-associated genes. Reduced hCINAP abundance impairs tumorigenesis, suggesting that hCINAP could serve as a diagnostic marker and chemotherapeutic target.

## Results

**CINAP deletion results in embryonic lethality in mice.** To determine the significance of hCINAP in higher multicellular species, the mouse homologue of hCINAP, designated CINAP, was deleted by homologous recombination. The portion of CINAP containing exons 3 and 4 was replaced with a cassette containing a neomycin resistance gene (Fig. 1a). The targeting vector was transfected into C57BL/6 mouse embryonic stem cells by electroporation. After G418 selection, 17 positive clones were identified by Southern blotting. Eight of the 17 positive clones were expanded for injection of BALB/C blastocysts. The chimeric mice were then crossed with C57BL/6J mice to obtain F1 mice carrying the recombined allele containing the floxed CINAP allele and Neo selection cassette. F1 mice were generated, after which genotyping was performed with the indicated primers (Supplementary Fig. 1a and Supplementary Table 1a). Female homozygous floxed CINAP mice were mated with male X-linked CMV-Cre mice to generate $CINAP^{+/-}$ mice with disrupted expression of CINAP exons 3 and 4, as well as expression of the Neo cassette (Fig. 1b). Female $CINAP^{+/-}$ mice were obtained (Supplementary Fig. 1b and Supplementary Table 1b) and intercrossed to generate $CINAP^{-/-}$ mice. Intercrossing of CINAP heterozygous mice produced heterozygous and wild-type mice with an approximate ratio of 2:1; however, no $CINAP^{-/-}$ offspring was obtained (>100 mice analysed) (Fig. 1c). Heterozygous $CINAP^{+/-}$ mice were viable and CINAP expression in different organs was evaluated by real-time quantitative PCR. The results showed that the CINAP mRNA levels of the organs of the $CINAP^{+/-}$ mice were reduced in comparison with those of the wild-type mice (Fig. 1d). Therefore, $CINAP^{+/+}$ and $CINAP^{+/-}$ mice were used in further studies to explore the role of CINAP.

**Reduced CINAP blocks 18S rRNA cleavage and rapid cell growth.** We performed northern blot assay using 5′-ITS1 and 5′-ITS2 probes (Fig. 2a) to detect whether CINAP heterozygotes show rRNA processing defects due to the essential role of hCINAP homologues in regulating 18S rRNA processing. Specific accumulation of 18S-E pre-rRNA was observed in liver and kidney tissue samples from $CINAP^{+/-}$ mice (Fig. 2b and Supplementary Fig. 2), indicating that the last step of 18S rRNA processing was blocked. As protein synthesis is associated with proper rRNA maturation, translation efficiency was evaluated in mouse embryonic fibroblasts (MEFs) from $CINAP^{+/+}$ and $CINAP^{+/-}$ mice. As shown in Fig. 2c, CINAP reduction decreased protein synthesis (0.72-fold); however, cell proliferation was not obviously delayed in $CINAP^{+/-}$ MEF cells (Fig. 2d). The weight of the $CINAP^{+/-}$ mice was similar to their wild-type littermates (Fig. 2e). These results suggest that 18S rRNA processing defects induced by CINAP reduction do not significantly impair normal cell proliferation and mouse development.

We hypothesized that reduced CINAP expression might act as a rate-limiting factor during rapid cell growth, as fast-growing cells are more sensitive to ribosome assembly defects. Therefore, insulin was used as the cell growth signal, to stimulate protein synthesis. Exposure to insulin induced $^{35}$S-methinoine incorporation by $CINAP^{+/+}$ MEF cells (1.26-fold), whereas insulin did not affect $^{35}$S-methinoine uptake by $CINAP^{+/-}$ MEF cells (Fig. 2f). These results demonstrate that CINAP haploinsufficiency limits the rate of protein synthesis during rapid growth. As cancer cells are characterized by rapid cell proliferation, we assessed the effect of hCINAP depletion on 18S rRNA processing and cell growth in cancer cells. hCINAP expression was knocked down by transfection with lentivirus expressing hCINAP-short hairpin RNA (shRNA). As shown in Fig. 2g, mRNA and protein levels of hCINAP in cancer cells harbouring hCINAP-shRNA-1

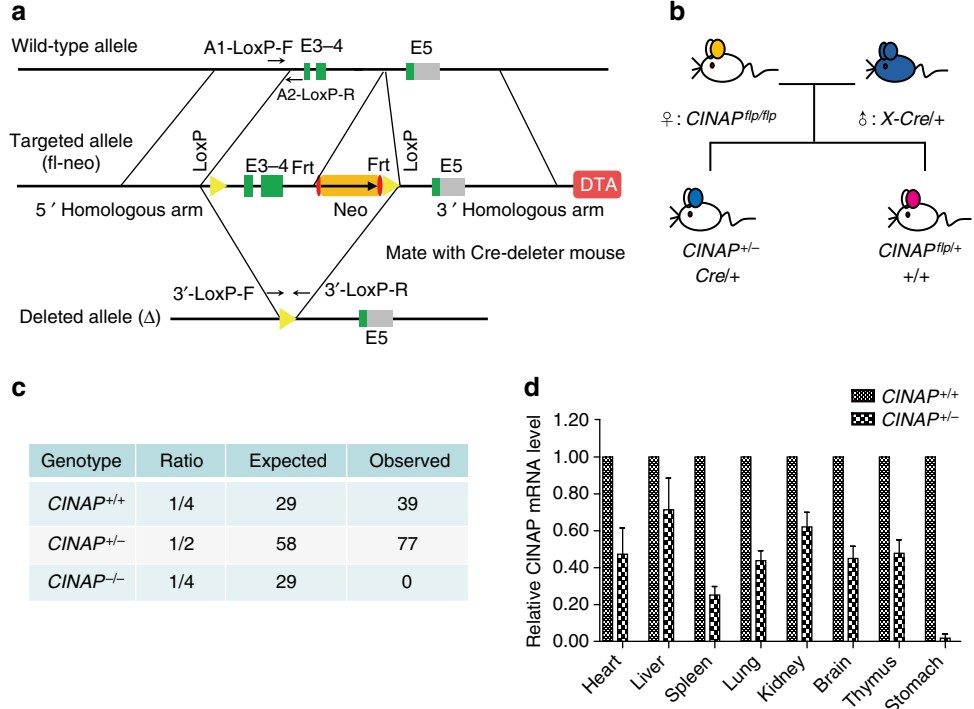

**Figure 1 | Disruption of CINAP results in embryonic lethality. (a)** Schematic diagram of the *CINAP* locus and the targeting construct. The targeting vector replaced exons 3 and 4 with a neomycin-selectable marker flanked by *LoxP* sites. **(b)** *CINAP^flp/flp^* mice (8 weeks of age, female, *n* = 2) were crossed with X-linked CMV-Cre mice (BALB/c, 8 weeks of age, male, *n* = 1) to generate *CINAP*-deleted mice. **(c)** Observed and excepted birth ratio from *CINAP^+/−^* intercrosses (*n* = 116). **(d)** Quantitative PCR analysis of *CINAP* expression in organs from adult *CINAP^+/+^* and *CINAP^+/−^* mice (female, 8 week of age, *n* = 3 for each genotype, randomly selected). Results are presented as mean ± s.d. Killing of mice was performed according to the ethical guidelines approved by Peking University Laboratory Animal Center.

or hCINAP-shRNA-2 were decreased to half of the corresponding levels in control cells, producing an effect similar to that of CINAP reduction in mice. Northern blot assay was performed to screen which 18S rRNA processing step was specifically affected by hCINAP. Consistent with the role of CINAP in mice, hCINAP depletion caused an obvious accumulation of the 18S-E pre-rRNA and a decreased level of matured 18S rRNA (Fig. 2h,i). Ribosome profiling assay showed that hCINAP reduction decreased the abundance of free 40S subunits, 80S ribosomes and polysomes (Fig. 2j). Consistently, hCINAP depletion significantly inhibited global protein synthesis in MCF7 cells (0.65-fold; Fig. 2k). MTT (3-(4,5-dimethylthiazol-2-yl)-2,5-diphenyltetrazolium bromide) assays showed that depletion of hCINAP significantly inhibited cancer cell growth (Fig. 2l). These results demonstrate that the defects of 18S rRNA processing and protein synthesis by hCINAP depletion are strikingly amplified in cancer cells and limit cancer cell proliferation.

**Role of the ATPase activity of hCINAP in 18S rRNA cleavage.** The essential role of hCINAP in 18S rRNA processing encouraged us to explore the underlying molecular mechanism. hCINAP is an adenylate kinase with high intrinsic ATPase activity[21,28]. To further assess the involvement of the enzymatic activity of hCINAP in 18S rRNA processing, northern blotting and ribosome profiling were performed with the enzymatic mutants hCINAP-D77G and hCINAP-H79G. The results showed that hCINAP depletion decreased 18S rRNA production and inhibited 40S subunit assembly. Wild-type hCINAP rescued the processing defects in hCINAP-depleted cells, whereas ATPase-defective hCINAP-H79G or D77G only slightly rescued production of 18S rRNA and 40S subunits (Fig. 3a and Supplementary Fig. 3a).

To investigate whether the enzymatic site mutation changed the conformation of hCINAP, we determined the crystal structure of hCINAP-D77G (Table 1, PDB No. 5JZV). Comparison of the crystal structure of hCINAP-D77G with that of wild-type hCINAP showed that no significant conformation occurred in the substrate-binding pocket of hCINAP-D77G; only a subtle conformational difference in the NMP-binding domain with a 0.354 Å root mean square deviation was observed (Fig. 3b). Stereo views of a portion of the $2F_o$-$F_c$ electron density maps and $F_o$-$F_c$ OMIT maps were shown in Supplementary Fig. 3b–d. These results suggest that replacement of a single residue of hCINAP does not cause significant rearrangement.

To distinguish whether the ATPase or adenylate kinase activity is responsible for the role of hCINAP in 18S rRNA processing, cells were treated with AP5A ((Di(adenosine-5′-)penta-phosphate), Tri-Li), a specific inhibitor of adenylate kinase. AP5A treatment did not significantly alter 18S rRNA abundance (Supplementary Fig. 3e), suggesting that the ATPase activity of hCINAP is involved in 18S rRNA maturation. Mass spectrometry analysis showed that RPS14, another protein functions in 18S rRNA processing, might bind with hCINAP (Supplementary Fig. 3f). Here we demonstrated that hCINAP bound to the carboxy-terminal tail of RPS14 (Fig. 3c). We assessed whether hCINAP blocked the binding between RPS14 and 18S-E pre-rRNA, because the C-terminal of RPS14 is involved in its binding with 18S-E pre-rRNA[29]. The addition of hCINAP decreased the binding affinity of RPS14 with premature 18S rRNA (Fig. 3d).

The functional characterization of hCINAP–RPS14 interaction also spurred us to determine whether RPS14 functions as an ATPase regulator to orchestrate the role of hCINAP in 18S rRNA processing. Thin-layer chromatography was performed with γ-[^32P]ATP as the ATPase substrate. Interestingly, the ATPase

activity of hCINAP was strongly stimulated by RPS14 (Fig. 3e), suggesting that RPS14 regulates the role of hCINAP in 18S rRNA processing by acting as an ATPase-activating protein. Next, we determined whether ATP binding and ATP hydrolysis by hCINAP regulate hCINAP–RPS14 interaction. *In vitro* pull-down assay was performed with GST-RPS14 and His-hCINAP in the presence of ATP or non-hydrolysable ATP analogue

AMP-PNP. As shown in Fig. 3f, the association of RPS14 with hCINAP was decreased by the addition of ATP, but not by AMP-PNP. This result suggests that ATP hydrolysis by hCINAP modulates dissociation of the hCINAP–RPS14 complex.

To further demonstrate that hCINAP is directly involved in the 18S rRNA processing, immunoprecipitation (IP), reverse transcriptase–PCR and electrophoretic mobility shift assay were

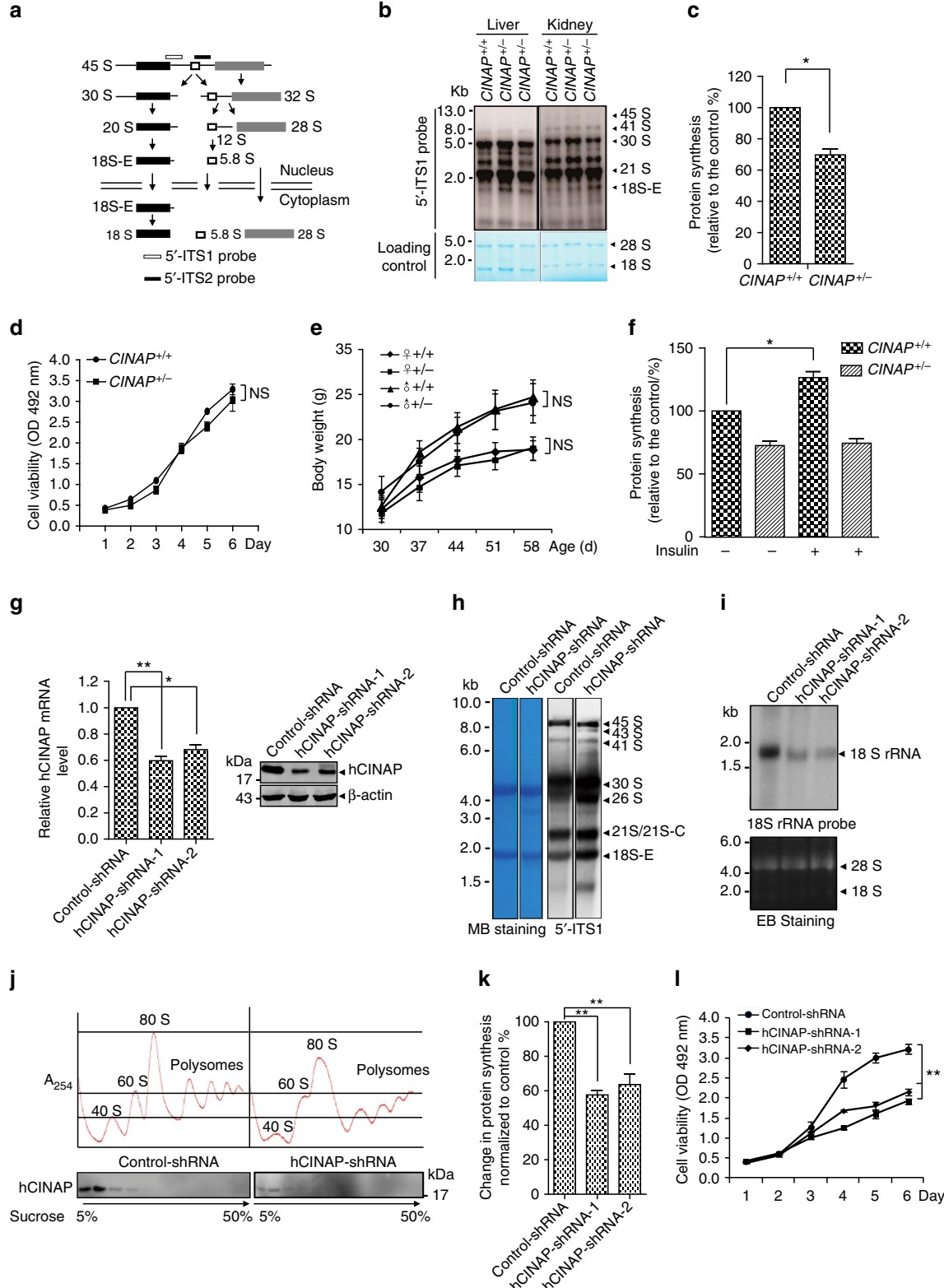

first performed and the results showed that hCINAP binds to 18S-E pre-rRNA both *in vivo* and *in vitro* (Supplementary Fig. 3g,h). As no known nuclease motifs are evident in the amino acid sequence of hCINAP, it seems unlikely to be that hCINAP functions as an endonuclease catalysing 18S-E pre-rRNA cleavage after dissociation with RPS14. Endonuclease Nob1 is responsible for generating the 3'-end of 18S rRNA in yeast and humans[30–33]. To assess whether hCINAP regulates Nob1 activity, we first detected the binding between hCINAP and Nob1. *In vitro* pull-down assays showed that hCINAP directly interacted with Nob1 (Fig. 3g). Co-IP assay further showed that the Zn-binding domain of Nob1 mediated its interaction with hCINAP (Supplementary Fig. 3i). To study the direct involvement of hCINAP in Nob1-mediated 18S rRNA processing, the efficiency of 18S-E pre-rRNA cleavage by Nob1 was evaluated using an *in vitro* RNA cleavage assay. Nob1 alone showed low nuclease activity, whereas addition of wild-type hCINAP but not the enzymatic mutants strongly stimulated 18S-E pre-rRNA cleavage by Nob1 (Fig. 3h and Supplementary Fig. 3j,k). Collectively, these data demonstrate that hCINAP functions as an ATP-regulated switch that interacts with RPS14 and Nob1, and is required for Nob1-mediated 18S rRNA processing (Fig. 3i).

**hCINAP reduction impairs tumour growth.** We assessed whether downregulation of hCINAP blocks tumorigenesis because of the essential role of hCINAP in Nob1-mediated 18S rRNA processing and the inhibitory effect of hCINAP depletion on ribosome assembly in human cancer cells. Fluorescence-activated cell sorting (FACS) analysis showed that hCINAP depletion induced cell cycle arrest at G1 phase (Fig. 4a,b). hCINAP-depleted HCT116 cells also showed a significantly increased apoptosis rate in comparison with that of the control group (Fig. 4c,d). Consistently, soft agar colony-formation assay showed that hCINAP-depleted HCT116 cells produced smaller colonies in comparison with wild-type cells (Fig. 4e). To detect hCINAP depletion-induced cell growth inhibition is due to its function in 18S rRNA processing, 18S rRNA rescue experiment was performed by transfecting 18S rRNA into hCINAP-depleted cells. The results showed that rescue of 18S rRNA significantly recovered cell growth (Supplementary Fig. 4a,b). Consistently, depletion of hCINAP and Nob1 caused a similar cell growth inhibition (Supplementary Fig. 4c). As hCINAP and Nob1 both function in 18S rRNA maturation, defects of cell growth by hCINAP depletion is mainly through regulating 18S rRNA processing even though other mechanisms may also be involved.

As the critical role of hCINAP in regulating tumour cell growth, to determine whether hCINAP is required for tumorigenesis *in vivo*, HCT116 cells with stably expressed hCINAP-shRNA or control-shRNA were injected subcutaneously into the right flank of nude mouse, after which tumour growth was monitored for 3 weeks. Injection of cells with control-shRNA generated large tumours (Fig. 4f, upper and middle panel). In contrast, hCINAP-depleted cells did not result in tumour generation (Fig. 4f, lower panel). To further provide *in vivo* evidence for the role of hCINAP in tumorigenesis, $CINAP^{+/+}$ and $CINAP^{+/-}$ mice were crossed with MMTV-polyomavirus middle T antigen (PyMT) transgenic mice, which have been widely used in mammary tumorigenesis study. The result showed that all the female $PyMT/CINAP^{+/+}$ mice spontaneously developed breast tumours at 70 days of age, whereas the $PyMT/CINAP^{+/-}$ mice started developing tumours at 90 days and only 40% (2/5) of the mice generated tumours at 120 days (Supplementary Fig. 4d). These data indicate that CINAP reduction inhibits tumorigenesis.

**hCINAP is highly expressed in human cancers.** To further illuminate the role of hCINAP in human cancers, we measured hCINAP expression in a panel of human breast cell lines. hCINAP showed higher expression in breast cancer cell lines than in normal breast cells (Fig. 5a). Similar results were obtained in other cancer cell types (Supplementary Fig. 5a). To compare *hCINAP* mRNA levels in tumour tissue and normal tissue, we collected 20 paired samples of breast cancer tissue and matched adjacent normal tissue. *hCINAP* mRNA expression was significantly upregulated in human breast tumours in comparison with that of normal tissue (Fig. 5b). To further confirm the high expression level of hCINAP in cancers, hCINAP protein levels in paired samples from 31 breast cancer patients and 90 colorectal adenocarcinoma patients were assessed by immunohistochemistry (IHC) (Supplementary Fig. 5b,c and Supplementary Data 1). Endogenous hCINAP is observed both in the cytoplasm and nucleus (Supplementary Fig. 5d,e). A scoring system was used to quantify the data from IHC tissue arrays and the results showed that in cancer tissue, the expression level of hCINAP in the cytoplasm but not in the nucleus was significantly higher than that of adjacent non-cancerous tissue (Fig. 5c–f and Supplementary Fig. 5f).

To evaluate the clinical significance of hCINAP, we examined the percentage of cancer patients with highly expressed hCINAP. Significantly increased hCINAP expression was found in 83.87% of breast tumour samples and 82.22% of colorectal adenocarcinoma samples (Fig. 5g). Intriguingly, patients with colorectal adenocarcinoma bearing tumours with high level of hCINAP showed poor overall survival than those bearing tumours with low level of hCINAP (Fig. 5h). These results indicate that hCINAP has considerable clinical significance and can be regarded as a particularly useful biomarker for cancer diagnosis.

**Figure 2 | CINAP deletion inhibits 18S rRNA processing and blocks rapid cell growth.** (**a**) Schematic diagram of rRNA processing in eukaryotes. The 5'-ITS1 and 5'-ITS2 probes used in the northern blot assay are indicated. (**b**) Total RNA of organs from $CINAP^{+/+}$ and $CINAP^{+/-}$ mice (female, 6 week of age, 3 mice used, randomly selected) was extracted and subjected to northern blot analysis with the 5'-ITS1 probe. 28S and 18S rRNA were shown as loading control. Killing of mice was approved by Peking University Laboratory Animal Center. (**c**) $^{35}$S-methionine labelling of $CINAP^{+/+}$ and $CINAP^{+/-}$ MEF cells. The number of counts was normalized to protein content. *$P < 0.05$ (Student's $t$-test). (**d**) Cell proliferation rate of $CINAP^{+/+}$ and $CINAP^{+/-}$ MEF cells was measured by MTT assays. (NS, no significance, two-way analysis of variance (ANOVA)). (**e**) The body weight of female and male mice (10 mice per group, randomly assigned) was recorded weekly from 30 to 60 days after birth. Results are presented as mean ± s.d. (no significance, two-way ANOVA). (**f**) $^{35}$S-methionione labelling in MEF cells stimulated with 100 nM insulin. The methionine incorporation level of wild-type MEF cells without insulin stimulation was considered as 100%. *$P < 0.05$ (Student's $t$-test). (**g**) hCINAP knockdown efficiency was assessed by quantitative reverse transcriptase–PCR and western blotting. *$P < 0.05$ and **$P < 0.01$ (Student's $t$-test). (**h**) Northern blot analysis with the 5'-ITS1 probe was performed to detect the accumulation of 18S rRNA precursors with hCINAP depletion. 28S and 18S rRNA were shown as the loading control. (**i**) 18S rRNA production was detected by northern blotting using 18S rRNA probe with hCINAP reduction. (**j**) Ribosome profiling was performed with MCF7 cells expressing control-shRNA or hCINAP-shRNA. Formation of 40S subunit was monitored by measuring the absorbance at 254 nm. Presence of hCINAP in each fraction was detected by western blot using anti-hCINAP antibody. (**k**) HeLa cells harbouring control-shRNA or hCINAP-shRNA were incubated with 1 μCi per ml $^{35}$S-Methionine and the incorporation of $^{35}$S-methionione was detected. **$P < 0.01$ (Student's $t$-test). (**l**) Cell proliferation assay was performed with HCT116 cells expressing control-shRNA or hCINAP-shRNA. Results (mean ± s.d) were analysed by two-way ANOVA.

**Highly expressed hCINAP promotes cancer cell growth**. We determined the effect of hCINAP overexpression on cell proliferation as hCINAP is highly expressed in cancers. MTT assays showed that U2OS cells harbouring high levels of hCINAP had a higher proliferation rate than that of the control cells (Fig. 6a). As shown in Fig. 6b, U2OS cells with stable hCINAP expression showed a decreased proportion of cells in the G0/G1 phase in comparison with that of the control cells. In addition, to evaluate the effect of hCINAP overexpression on cell apoptosis, cisplatin ($10 \mu g \, ml^{-1}$), a chemotherapy drug, was used to induce cell apoptosis. The result showed treatment of cisplatin jumped the cell apoptosis and overexpression of hCINAP significantly resisted cisplatin-induced cell apoptosis (Fig. 6c,d).

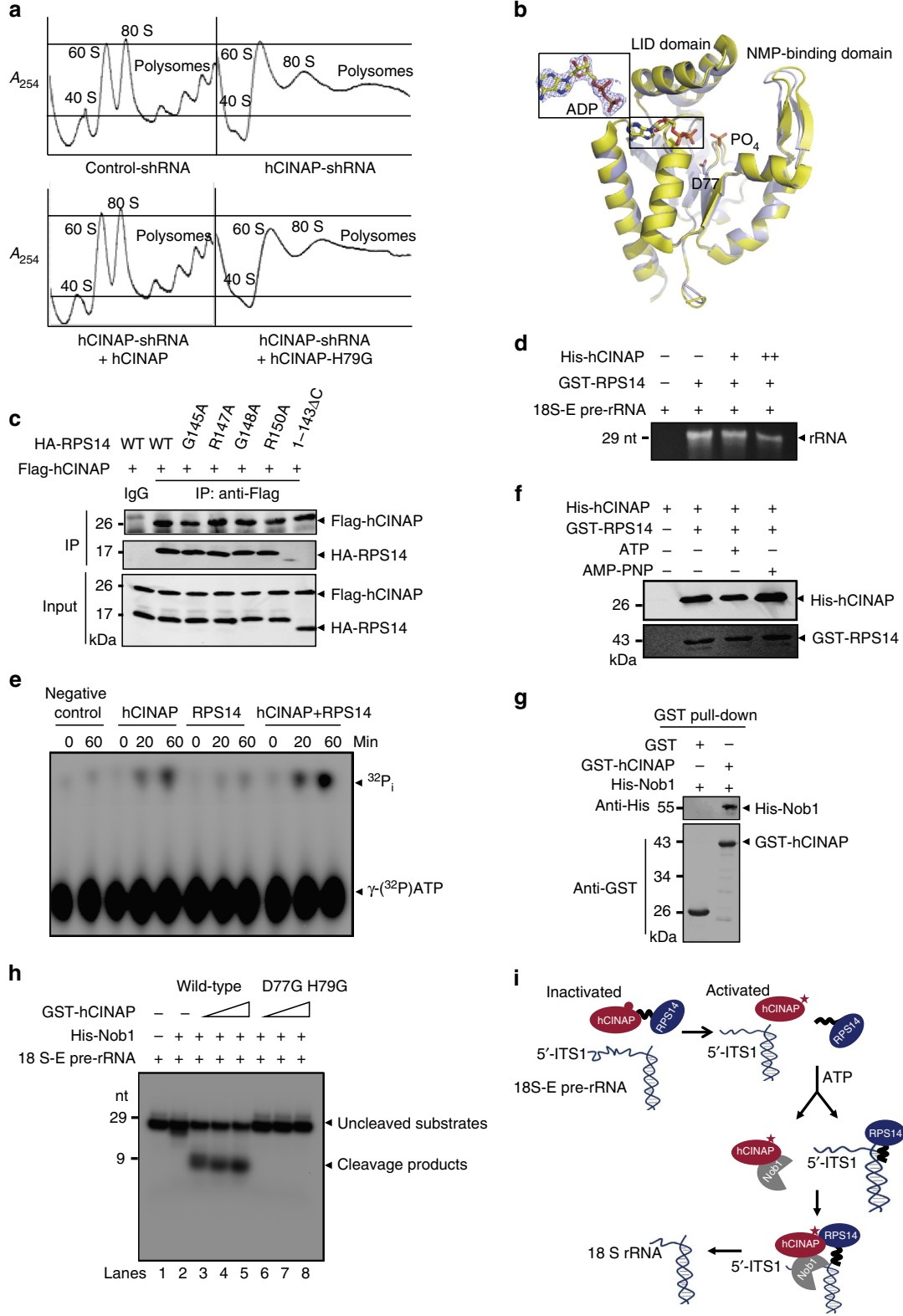

To examine the effect of highly expressed hCINAP on tumour growth, we performed *in vitro* clone-formation assay. As shown in Fig. 6e, U2OS cells with highly expressed hCINAP produced more and larger clones than that of the control cells. Consistently, nude mice tumorigenesis assay showed the tumour weight of the nude mice injected with U2OS cells with overexpressed hCINAP were significantly increased to threefold in comparison with that of U2OS cells harbouring the control vector (Fig. 6f,g). These results demonstrated that highly expressed hCINAP promotes tumour growth.

**Upregulated hCINAP promotes cancer-related mRNA translation**. Increased protein synthesis and altered protein synthesis patterns are associated with cell transformation. We assessed whether highly expressed hCINAP in cancers drives protein synthesis of factors involved in tumorigenesis, because hCINAP is required for ribosome assembly and promotes cancer cell growth. To assess global changes in mRNA translation, RNA sequencing (RNA-seq) on polysome profiling was performed with MCF7 cells by sucrose-gradient centrifugation, to determine the mRNA distribution in each ribosome fraction (Fig. 7a). The position of an mRNA within the sucrose gradient reflected its translational status. Actively translated mRNA was recruited to the polysomes[34]. Ribosome profiling revealed that hCINAP overexpression led to increased polysome abundance (Fig. 7b). Compared with the control cells, hCINAP overexpression was associated with a relative increase of $>16$-fold in the mRNA abundance of polysomes to monosomes (Fig. 7c). To define the translational efficiency of each mRNA, mRNAs corresponding to monosome and polysome fractions were isolated and subjected to RNA-seq. The translational efficiency of significantly upregulated or downregulated mRNAs was assessed (Supplementary Data 2). As shown in Fig. 7d, $\sim 1,722$ mRNAs were significantly recruited onto the polysome, 1,521 mRNAs were released off the polysome and $\sim 5,000$ mRNAs were unchanged with the highly expressed hCINAP.

To illuminate the physiological processes that these upregulated and downregulated mRNAs involved in, the mRNAs were examined for enrichment in genes associated with Kyoto Encyclopedia of Genes and Genomics pathways. Kyoto Encyclopedia of Genes and Genomics pathways exhibiting term enrichment at a significant level of $P<0.05$ are shown in Fig. 7e. mRNAs associated with cancer-related pathways showed substantial enrichment. We confirmed the polysome RNA-seq data by quantifying the mRNA relative abundance of the top ranked oncogenes and tumour suppressors in monosomes and polysomes (Fig. 7f). With overexpression of hCINAP, mRNAs encoding oncogenes Myc, SNCG (γ-synuclein), which increases metastasis and promotes genetic instability in breast cancer,

CASC5 (cancer susceptibility candidate 5), as well as VOPP1 (vesicular overexpressed in cancer, prosurvival protein 1) were strongly shifted to the polysome fractions, indicating active translation (Fig. 7f). In addition, the BRCA1 (breast cancer susceptibility gene 1) was shifted to the monosome fractions, suggesting translation inhibition (Fig. 7f).

To further understand the role of hCINAP in regulating cancer cell growth, we performed Connectivity Map analysis with translation, significantly changing mRNAs to discover the connections between drugs and gene expression. hCINAP overexpression-induced gene expression signature showed a strongly negative correlation with the results obtained by two anti-cancer drugs, verteporfin and lycorine (Supplementary Data 3). Verteporfin was exhibited to inhibit cancer progression partially by impairing the global clearance of high molecular weight of oligomerized proteins including p62 and STAT3 (ref. 35). Lycorine has an inhibitory effect on proliferation and invasion of leukaemia cells[36]. These results suggested a possible cross-talk

**Table 1 | Data collection and refinement statistics of hCINAP-D77G.**

|  | hCINAP-D77G |
| --- | --- |
| *Data collection* |  |
| Space group | P 6₁ |
| Cell dimensions |  |
| *a, b, c* (Å) | 101.76, 101.76, 58.85 |
| α, β, γ (°) | 90, 90, 120 |
| Resolution (Å) | 50–2.07 (2.14–2.07)* |
| $R_{merge}$ (%) | 7.4 (48.6) |
| $I/\sigma I$ | 50.2 (8.1) |
| Completeness (%) | 100 (100) |
| Redundancy | 22.9 (21.7) |
|  |  |
| *Refinement* |  |
| Resolution (Å) | 44.06–2.07 |
| No. reflections | 21343 |
| $R_{work}/R_{free}$ | 0.190/0.235 |
| No. atoms |  |
| Protein | 1,438 |
| Ligand (ADP) | 27 |
| Water | 171 |
| B-factors (Å²) |  |
| Protein | 32.6 |
| Ligand (ADP) | 49.8 |
| Water | 46.2 |
| Root mean square deviations |  |
| Bond lengths (Å) | 0.033 |
| Bond angles (°) | 2.019 |

*Highest resolution shell is shown in parenthesis.

**Figure 3 | The ATPase activity of hCINAP is orchestrated by RPS14 and essential for Nob1-mediated 18S rRNA processing.** (**a**) hCINAP-depleted cells were transfected with wild-type or ATPase-defective hCINAP, after which 40S subunit formation was detected by sucrose density gradient centrifugation. (**b**) Structural superposition of hCINAP-D77G (yellow) with hCINAP-ADP (light blue, PDB code: 5JZV). The $2F_o–F_c$ map of ADP in the structure of hCINAP-D77G-ADP is contoured at 1.4 ó (shown in the box). The side chain of D77 and a phosphate group in hCINAP-ADP are also shown. (**c**) 293T cells were transfected with Flag-hCINAP and the mutants or truncation of HA-RPS14. Cells were harvested and the interaction between hCINAP with RPS14 was examined with indicated antibodies. (**d**) Pull-down assay was carried out to detect the effect of hCINAP on the binding capacity of RPS14 with 18S-E pre-rRNA. (**e**) Thin-layer chromatography analysis was performed with γ-[³²P]ATP to examine the effect of RPS14 on the ATPase activity of hCINAP. Radioactively labelled products were visualized using a phosphor screen. (**f**) *In vitro* pull-down assay was performed to detect the interaction between hCINAP with RPS14 in the presence of ATP or non-hydrolysable ATP homologue AMP-PNP. (**g**) Determination of hCINAP and Nob1 interaction by GST pull-down experiments. (**h**) *In vitro* RNA cleavage assay to detect 18S-E pre-rRNA cleavage by Nob1 with wild-type hCINAP or the enzymatic mutants of hCINAP. (**i**) A model of regulation of 18S rRNA processing by hCINAP. In the pre-40S particles, hCINAP binds to the C-terminal of RPS14 and blocks binding of RPS14 to premature 18S rRNA, which may lead to a conformational change in the pre-rRNA. Meanwhile, RPS14 stimulates the ATPase activity of hCINAP, whereas ATP hydrolysis by hCINAP dissociates the hCINAP–RPS14 complex, which may facilitate proper assembly of RPS14 with rearranged pre-rRNA. Subsequently, hCINAP interacts with Nob1 and activates Nob1-mediated 18S rRNA processing.

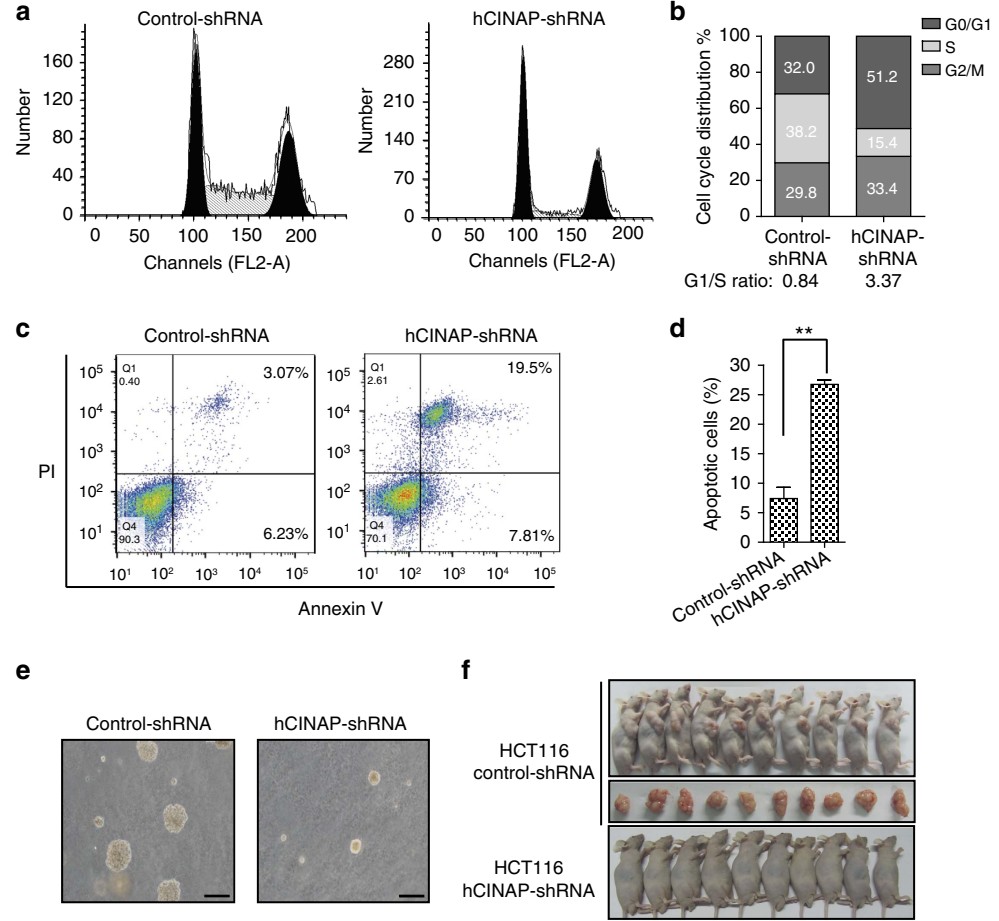

**Figure 4 | hCINAP depletion inhibits tumorigenesis. (a,b)** hCINAP depletion caused cell cycle arrest at the G1 phase. HCT116 cells stably expressing control-shRNA or hCINAP-shRNA were subjected to FACS analysis. The percentage of cells in each phase was obtained from three independent experiments. **(c,d)** Apoptosis was measured with cells labelling with AnnexinV–fluorescein isothiocyanate (FITC) and propidium iodide and the signal was obtained using a BD FACS Caliber instrument. Results are presented as mean ± s.d. **P < 0.01 (Student's t-test). **(e)** HCT116 cells with stably expressed control-shRNA or hCINAP-shRNA were seeded in the soft agar and cultured for 2 weeks. Representative images of colonies formed by each type of cells were shown. Scale bar, 200 µm. **(f)** Tumour formation in nude mice (female, BALB/c, 5 weeks of age, 10 mice per group, randomly assigned) injected subcutaneously with HCT116 cells harbouring control-shRNA or hCINAP-shRNA. Tumours were isolated after 3 weeks. This experiment was carried out according to the ethical guidelines and approved by the Peking University Laboratory Animal Center.

of hCINAP with the downstream signalling pathways of verteporfin and lycorine.

The translation efficiency of mRNA is regulated by its 5′-untransated region (5′-UTR) structure. To figure out whether specific motif(s) distribute in the mRNAs that were translationally upregulated by hCINAP, MEME motif elicitation software was used to seek the enriched motifs (up to 12 nt for motif length) in the 5′-UTR region of those RNAs. The most significantly enriched motif is G/C-rich motif in the form of $C(GGC)_3GG$ (Fig. 7g), which is highly suggestive of G-quadruplex formation. This result suggests that highly expressed hCINAP preferentially promotes translation of cancer-associated mRNA with specific GC-rich motif.

## Discussion

In the present study, we characterized hCINAP as an essential factor in embryogenesis and tumour growth that regulates ribosome assembly and controls translation. During normal cell growth, hCINAP regulates 18S rRNA processing through a hierarchical interaction between 18S-E pre-rRNA, RPS14 and Nob1. When malignancy occurs, highly expressed hCINAP promotes ribosome assembly and selectively modulates

translation of cancer-associated genes (Fig. 7h). Notably, the highly expressed hCINAP in cancer cells confirms the clinical significance of hCINAP. hCINAP depletion abolished tumorigenesis, indicating that hCINAP is an attractive therapeutic target for cancer treatment.

The process of ribosome biogenesis has been studied extensively in unicellular organisms. Studies in yeast and archaea have demonstrated that hCINAP homologues function in small ribosomal subunit assembly[22,27,37]. We advocated the study of ribosome biogenesis in multicellular organisms, because dysfunctional ribosome assembly is strongly associated with human genetic diseases and cancers. In this study, we demonstrated that hCINAP binds to the C-terminal RNA-binding element of RPS14. Recent studies showed that many ribosomal proteins contain universally conserved terminal extensions, which mediate the interactions of ribosomal proteins with the dedicated chaperones and transport systems, to assure the fail-safe targeting of ribosomal proteins to the assembly sites[38–40]. In our study, hCINAP chaperones RPS14 assembly into pre-40S particles through transient binding with RPS14, to inhibit premature interaction between RPS14 and 18S pre-rRNA, which is assumed to facilitate the conformational change of 18S pre-rRNA that allows proper incorporation of

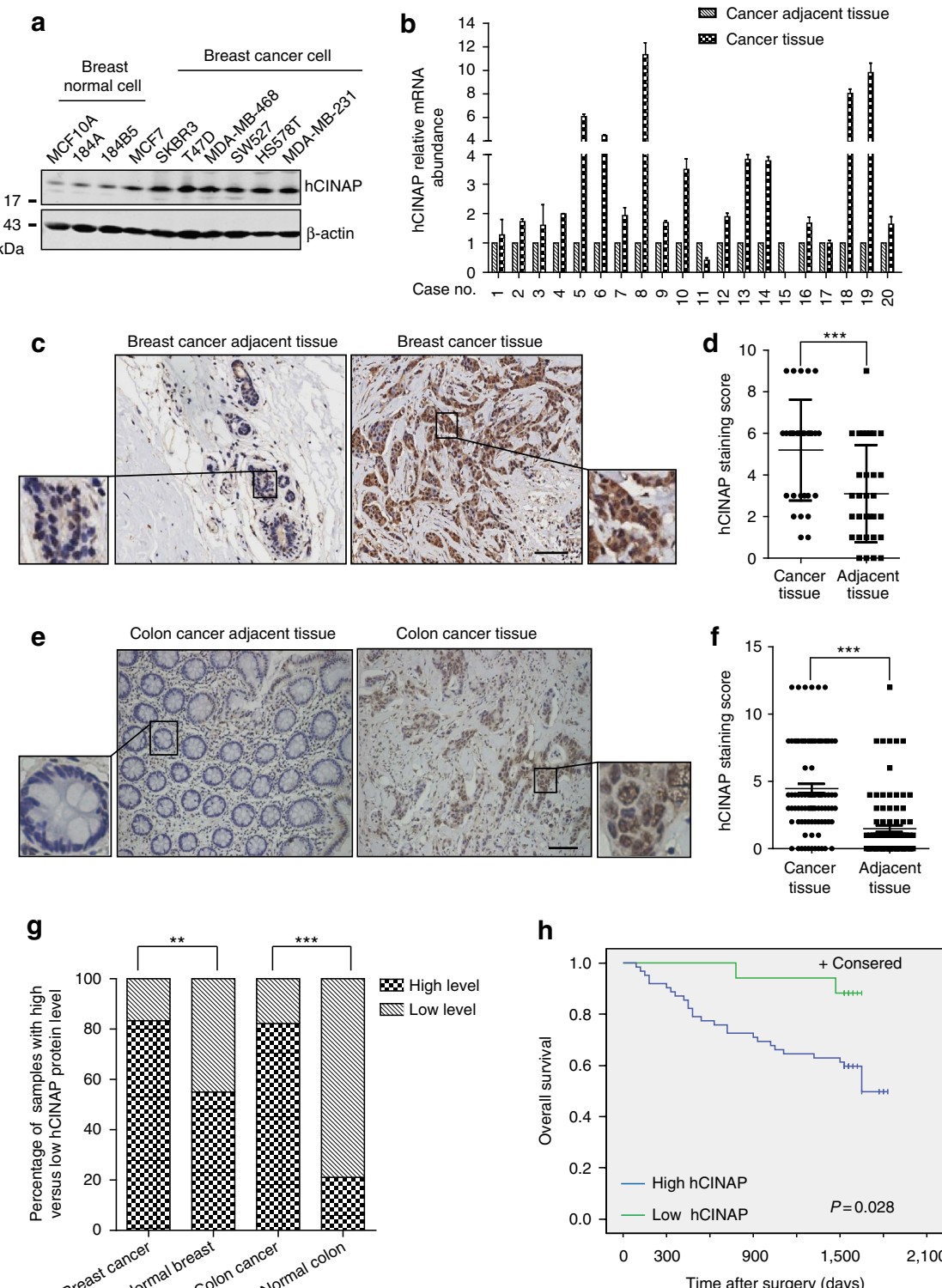

**Figure 5 | hCINAP is highly expressed in human cancers and correlated with a worse prognosis.** (**a**) Expression of hCINAP in a panel of human breast cell lines was detected by western blotting. (**b**) Quantitative reverse transcriptase–PCR was used to detect the relative mRNA level of *hCINAP* in human breast tissue, which was compared with that of adjacent paired cancer tissue. Results are presented as mean ± s.d. (**c,e**) IHC was performed using anti-hCINAP antibodies to detect hCINAP expression with human breast cancer and colorectal adenocarcinoma cancer tissue arrays with diaminobenzidine (DAB) staining. Scale bars, 50 μm. (**d,f**) Plots of the scores for the cytoplasmic hCINAP in each pair of tissue samples. Statistical significance was determined by the Mann–Whitney *U*-test. (**g**) The stacked bars indicate the percentages of samples with high and low hCINAP expression levels relative to the total number of samples in each section. Data were analysed using Pearson's $\chi^2$-test. **$P < 0.01$ and ***$P < 0.001$. (**h**) Kaplan–Meier curve of the 5-year overall survival of 90 patients with colorectal adenocarcinoma. Patients were divided into 'high hCINAP' group (the score of hCINAP expression in the cancer tissue $> 2$) and 'low expression' group (the score of hCINAP expression in the cancer tissue $\leq 2$). The difference of the overall survival between these two groups was determined using a log-rank test.

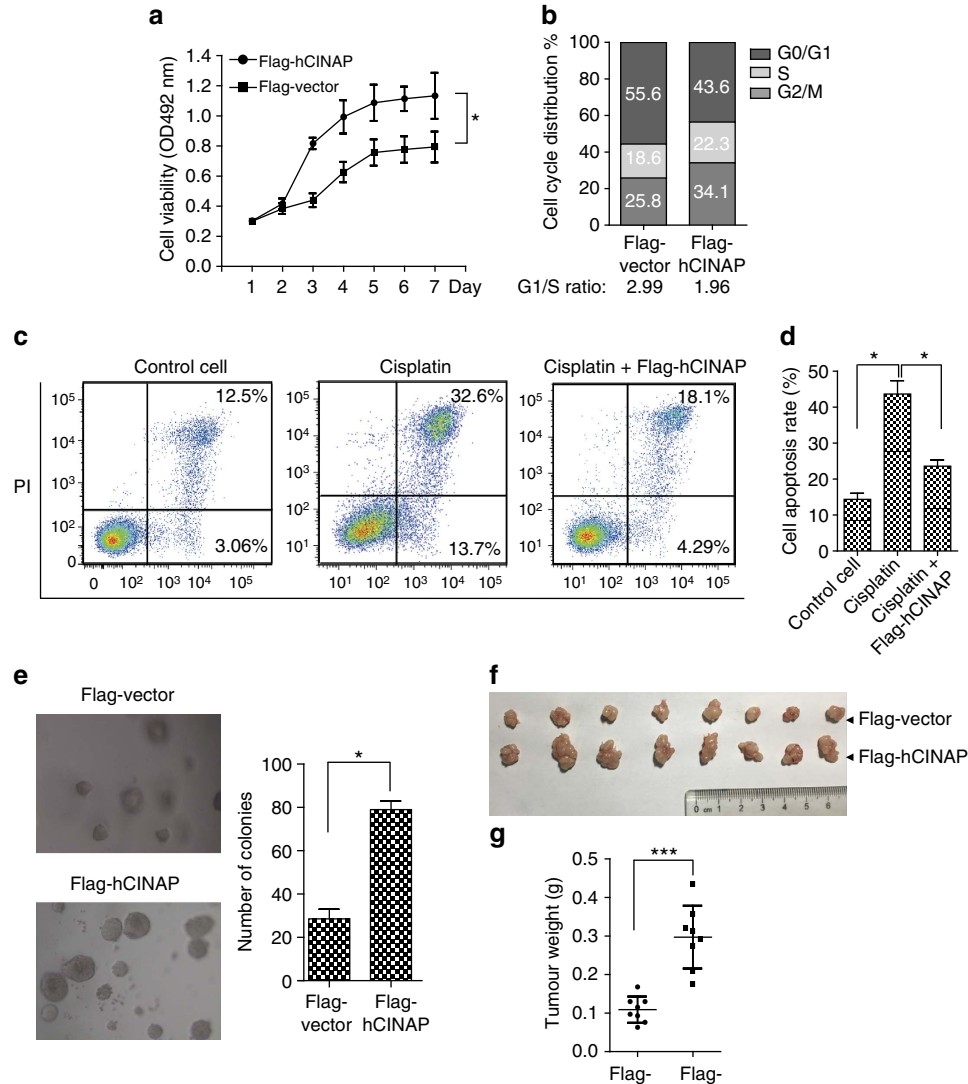

**Figure 6 | Highly expressed hCINAP promotes cancer cell growth.** (**a**) Growth of U2OS cells harbouring Flag-hCINAP or Flag-vector was monitored by MTT assay. Three independent experiments were performed. Results are presented as mean ± s.d. *$P < 0.05$ (two-way analysis of variance (ANOVA) test). (**b**) Cell cycle progression of U2OS cells with overexpressed hCINAP was measured using FACS analysis. (**c**,**d**) U2OS cells were subjected to cisplatin (10 μg ml$^{-1}$) treatment for 24 h. The effect of hCINAP overexpression on cisplatin-induced cell apoptosis was examined by AnnexinV–fluorescein isothiocyanate (FITC)/PI FACS analysis. Results are presented as mean ± s.d. *$P < 0.05$ (Student's t-test). (**e**) Soft agar colony-formation assay was performed by seeding U2OS cells with stably expressed Flag-hCINAP or control vector in the 96-well plate. The graph showed the number of colonies after 3 weeks. Results are presented as mean ± s.d. *$P < 0.05$ (Student's t-test). (**f**,**g**) U2OS cells with highly expressed hCINAP or control vector were injected into the flank of nude mice (BALB/c, female, 5 weeks of age, 8 mice, randomly assigned). After 3 weeks, tumours were isolated and tumour weight was evaluated. Results are presented as mean ± s.d. ***$P < 0.001$ (Student's t-test). This experiment was approved by Peking University Laboratory Animal Center.

RPS14. Interestingly, binding between hCINAP and RPS14 is dynamically regulated by ATP hydrolysis. When cellular ATP levels are relatively low, such as during nutrient deprivation, dissociation of the hCINAP–RPS14 complex may be blocked, inhibiting ribosome biogenesis and conserving energy. Working as a switch, ATP modulates dissociation of hCINAP and RPS14, and facilitates proper assembly of RPS14 and pre-ribosomal particles.

In this study, hCINAP was found to bind to Nob1 and the ATPase activity of hCINAP was demonstrated to be essential for triggering Nob1-mediated 18S-E pre-rRNA cleavage. Interestingly, yeast hCINAP homologue Fap7 does not interact with Nob1 (ref. 22). We speculated that hCINAP interacts with Nob1, on which ATPase hydrolysis by hCINAP may provide energy to

induce a structure rearrangement in Nob1, which may stimulate the endonuclease activity of Nob1. Therefore, solving the complex structure of hCINAP-Nob1 is quite important. Notably, different from the stimulating function of hCINAP in Nob1-mediated 18S rRNA maturation, PhFap7 was shown to inhibit PhNob1-mediated RNA cleavage[37]. Yeast lacking Fap7 are unable to induce cleavage of site D and show prematurely terminated 18S rRNA processing[22]. It is possible that the function of hCINAP in regulating Nob1-mediated 18S rRNA processing is different in archaeal and eukaryotic organisms. Additional studies using other species are necessary to clarify the functional relationship between hCINAP and Nob1. The present study provides important insight into the diverse mechanisms underlying ribosome biology among different species.

In recent years, translational control has been studied as a determinant of tumorigenesis and a prospectively targetable process[2]. Consistent with the promotion effects of hCINAP on 18S rRNA production (Supplementary Fig. 8f) and ribosome assembly, hCINAP preferentially upregulates the translational efficiency of G/C-rich mRNAs that encoding proteins involved in cancer-related signalling pathways. A recent study showed that eIF6 depletion significantly affected the translation efficiency of mRNA containing G/C-rich motif in their 5′-UTRs[41]. As no significant effect was observed with the protein abundance of eIF4E or the phosphorylation level of 4EBP1 by abnormal expression of hCINAP (Supplementary Figs 6h and 8e), alterations in translational pattern regulated by

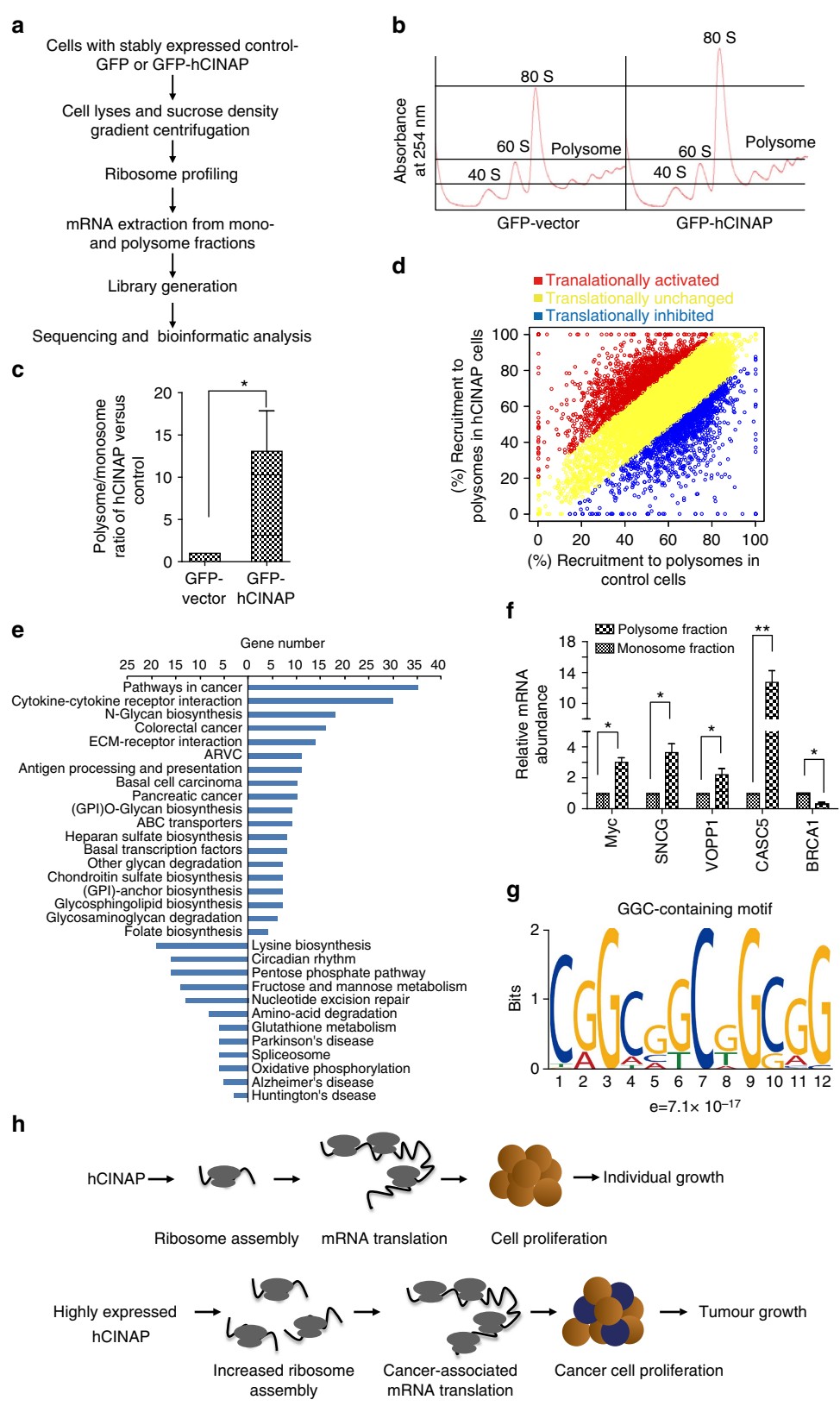

hCINAP is not associated with the signalling to translation apparatus but due to the effect of hCINAP on ribosome assembly. These results provide an important link between ribosome dysfunction and translation control. Recently, eIF6 depletion was shown to impair translation and caused coordinated changes of gene expression[41]. Beyond the translational regulation, highly expressed hCINAP induced a gene-expression signature marked by upregulating cation channel complex, sexual production and neurotransmitter metabolic process (Supplementary Fig. 7 and Supplementary Data 4). Future study may be desired to explore the function of hCINAP in these physiological processes and identify the coupling relationship of translation and transcription regulated by hCINAP.

Previous study demonstrated that hCINAP functions in Cajal body formation and hCINAP depletion in cancer cells reactivated p53 activity[26,42]. In this study, overexpression of hCINAP has no significant effect on Cajal body formation and p53 activity (Supplementary Fig. 8a–d). Depletion of hCINAP increased the protein abundance and activity of p53 without affecting p53 splicing (Supplementary Fig. 6a–c). It is documented that perturbations in rRNA processing lead to ribosome stress and activate p53 (ref. 43). The increases of p53 abundance and activity are most likely to be a result from the defects of ribosome assembly induced by depleted hCINAP. The inhibition of 18S rRNA processing and elimination of tumorigenesis by hCINAP depletion occurred in p53-deficient cells (Supplementary Fig. 6d–g), indicating that the roles of hCINAP in 18S rRNA maturation and cancer cell growth are independent from regulating p53.

hCINAP is highly expressed in cancers and a slightly increased expression pattern of hCINAP was observed with the progress of colon cancer (Supplementary Fig. 5g). A substantial increase in the number of samples is needed to accurately evaluate whether hCINAP expression is significantly correlated with the disease development in the future study. Given the important role of hCINAP in regulating cancer cell growth, knowledge of the transcriptional network regulating hCINAP expression should allow researchers to determine which oncogenes control hCINAP expression during tumorigenesis. It has been proposed that c-Myc acts as a master regulator of ribosome biogenesis and translational control by affecting transcription of many ribosomal proteins and ribosome assembly factors[12,44]. In our study, no significant regulation of hCINAP by c-Myc was observed. Nevertheless, the relatively high gene dosage of hCINAP in cancer cells suggests that human cancer cells may have a tendency to overexpress hCINAP. The remarkable inhibitory effect of hCINAP depletion on tumorigenesis demonstrates that exploration of upstream regulators of hCINAP and generation of selective small-molecule inhibitors targeting hCINAP may provide therapeutic strategies for patients with cancer.

## Methods

**Generation of CINAP knockout mice.** CINAP knockout mice were generated by homologous recombination. A homologous arm covering 5.2 kb upstream of CINAP exon 3 and 10.0 kb downstream of exon 4 was subcloned from a bacterial artificial chromosome clone isolated from a C57BL/6 mouse (BAC Genomic Library). An Frt-flanked Neo resistance-positive selection cassette was inserted downstream of exon 4, whereas loxP sites were introduced into intron 2 and intron 4. The negative selection marker DTA (Diphtheria Toxin Subunit A) was located downstream of the 3′-homologous arm. After linearization, the targeting vector was transfected into C57BL/6 embryonic stem cells by electroporation. After G418 selection, 17 positive clones were identified by southern blotting. Eight positive clones were injected into BALB/c blastocysts and implanted into pseudopregnant females. The chimeric mice were then crossed with wild-type C57BL/6J mice to obtain the F1 generation carrying the floxed CINAP allele and Neo selection cassette. Five F1 mice were generated and genotyped via PCR of tail-tip genomic DNA with the following primers: CINAP-A1-loxp-F, 5′-AGGGTGGCACATCCTGTAA TC-3′ and CINAP-A2-loxp-R, 5′-CGGCAACATGGCAACATAGC-3′. To generate CINAP knockout mice, CINAP-floxed mice were crossed with C57BL/6 CMV-Cre deleter mice. The following primers were used to detect deletion of the CINAP allele by PCR with Taq Plus Master Mix (Vazyme Code: P212-03, Vazyme Biotech Co., Ltd): CINAP-A1-loxp-F, 5′-AGGGTGGCACATCCTGTAATC-3′ and CINAP-A2-loxp-R, 5′-CGGCAACATGGCAACATAGC-3′; CINAP-3′-loxp-F, 5′-AGGGTGGCACATCCTGTAATC-3′ and CINAP-3′-loxp-R, 5′-CACGA-TAGTGTGGGATTTCTATCTGG-3′. All mice were housed at the Peking University Laboratory Animal Center. The experiments were performed following the 'Guide for the Care and Use of Laboratory Animals' and the 'Principles for the Utilization and Care of Vertebrate Animals'. All the animal work was approved by the Peking University Laboratory Animal Center.

**Plasmids and antibodies.** Expression plasmids of pRK-Flag-hCINAP and pRK-HA-RPS14 were constructed by inserting full-length complementary DNAs of hCINAP or RPS14 into the pRK vector at EcoRI and XbaI restriction sites. Insertions were verified by DNA sequencing. To generate recombinant lentivirus vectors expressing hCINAP-shRNA or non-silencing control-shRNA hCINAP-shRNA-1: 5′-CAGAGUAGUUGAUGAGUUA-3′, hCINAP-shRNA-2: 5′-GAGAG AAGGUGGAGUUAUU-3′ and non-silencing control-shRNA: 5′-UUCUCCGAA CGUGUCACGU-3′ were cloned into the pGCSIL-Puromycin lentivirus vector (GeneChem Co. Ltd, China). Antibodies including anti-Myc (M047-3), anti-GST (M071-3), anti-β-actin (PM053), anti-Lamin B1 (PM064), anti-His (D291-3) and anti-α-Tubulin (M175-3) were purchased from MBL. Monoclonal anti-Flag (M2; F3165) and anti-HA(HA-7; H9658) were from Sigma. Anti-p53 (DO-1) was from Cell Signaling Technology. IRDye 800CW Goat anti-mouse (926–32210) and IRDye 800CW Goat anti-rabbit (926–32211) were purchased from LI-COR Bioscience. Fluorescein isothiocyanate-conjugated goat anti-mouse IgG (ZF-0312) and TRITC-conjugated anti-rabbit IgG (ZF-0316) were from ZSGB-Bio. Rabbit polyclonal anti-hCINAP was generated as previously described[25].

**Cell lines and clinical samples.** HeLa, HCT116, MCF7 and U2OS cells (ATCC) were cultured in DMEM medium supplemented with 10% fetal bovine serum at 37 °C with 5% $CO_2$. MEF cells were obtained from embryos at E13.5D. Experiments were performed at passages 1 through 4. Human cancer samples were kindly provided by the Tissue Bank of Beijing Cancer Hospital. Informed consent was obtained from all the patients. The tissue collection procedure was approved by the Medical Ethics Committee of Beijing Cancer Hospital. The investigation was performed after approval by the Ethics Committee of Peking University.

**Western blot and northern blot analyses.** For western blotting, the proteins were separated with SDS–PAGE followed by transfer to the nitrocellulose filter membrane (PALL). The membrane was first blocked with 5% milk and then sequentially

**Figure 7 | High expressed hCINAP specifically promotes translation of mRNA involved in cancer-related signalling pathways.** (**a**) Work flow for RNA-seq on polysome profiling. Cells that stably expressed GFP–vector or GFP–hCINAP were subjected to sucrose density gradient centrifugation, to assess the distribution of monosomes and polysomes. RNA components corresponding to the monosome and polysome fractions were collected. A cDNA library was generated and RNA-seq was performed. (**b**) Representative ribosome profile of cells with hCINAP overexpression in comparison with that of the control cells. (**c**) The relative ratio of the RNA abundance in the polysome compared with that of monosome fractions. Results were obtained from three biological replicate experiments. Results are presented as mean ± s.d. P-values were calculated using Student's t-test (two-tailed). (**d**) The mRNA associated with polysomes in cells overexpressing hCINAP versus control were categorized as translationally inhibited (blue) and translationally activated (red) according to the polysome shift analysis. The cutoff for defining significant translational change was chosen when the percentage of polysome shift was 1 s.d. higher from the mean difference for all the identified mRNAs. The remaining unchanged mRNAs are represented in yellow. (**e**) Pathway enrichment analysis by DAVID showed that cancer-associated genes were predominantly translationally activated in cells that overexpressed hCINAP. (**f**) Polysome shift quantitative PCR analysis of the relative mRNA expression levels of Myc, SNCG, VOPP1, CASC5 and BRCA1 in the polysome fraction to that of monosome fraction. Results are presented as mean ± s.d. (**g**) MEME motif elicitation software was used to identify the enriched motifs (up to 12 nt for motif length) in the 5′-UTR region of the mRNAs with activated translation by hCINAP overexpression. (**h**) Working model for regulation of ribosome assembly and tumorigenesis by hCINAP. Under normal growth conditions, hCINAP is essential for ribosome biogenesis and cell proliferation. During malignant transformation, hCINAP overexpression promotes ribosome assembly and specifically modulates translation of cancer-associated genes to facilitate cancer cell growth.

incubated with the indicated primary antibodies. For the northern blotting, total RNA was isolated using TRIZOL Reagent (Invitrogen). RNA (5 µg) was separated on a 1% agarose-formaldehyde gel and transferred to Hybond N$^+$ membranes (Amersham). The probes were labelled with γ-[$^{32}$P]ATP using the QIAquick Nucleotide Removal Kit (Qiagen) according to the manufacturer's protocols. The following oligonucleotides were used: 5′-ITS1 (mouse), 5′-ACgCCgCCgCTCCTCC ACAgTCTCCCgTT-3′, 5′-ITS2 (mouse), 5′-ACTggTgAggCAgCggTCCgggAggCg CCgACg-3′ and 5′-ITS1(human), 5′-CCTCgCCCTCCgggCTCCgTTAATgATC-3′, 18S rRNA (human), 5′-gCATggCTTAATCTTTgAgCAAgCATAT-3′. Uncropped scans of the blottings are presented as Supplementary Fig. 9.

**Measurement of protein synthesis by $^{35}$S-methinoine labelling.** For *in vivo* labelling, cells were incubated with 3 µCi ml$^{-1}$ $^{35}$S-methionine (Amersham Pharmacia Biotech) for 1 h after starvation in methionine-free DMEM. Cells were harvested by scraping and centrifuged at 830 g for 5 min at 4 °C, after which the cell pellets were washed with pre-chilled PBS and lysed with RIPA buffer (0.05 mM Tris-HCl pH 7.2, 0.15 mM NaCl, 1% Triton X-100, 1% sodium deoxycholate and 0.1% SDS) for 30 min on ice. The supernatant was precipitated using tri-chloroacetic acid and filtered with glass fibre discs under vacuum. The signal was analysed using a liquid scintillation counter (Tri-Carb 2100TR, Perkin Elmer).

**Ribosome profile.** Cells were treated with 100 µg ml$^{-1}$ cycloheximide (CHX) for 10 min, washed with pre-chilled PBS containing 100 µg ml$^{-1}$ CHX and then resuspended in hypotonic buffer (10 mM HEPES pH 7.9, 1.5 mM MgCl$_2$, 10 mM KCl, 0.5 mM dithiothreitol (DTT), 100 µg ml$^{-1}$ CHX, 40 U ml$^{-1}$ RNase inhibitor and 1 × Protease Inhibitor Cocktail) on ice for 15 min. Samples were mechanically disrupted (15 strokes) with a Dounce homogenizer (KONTES). The resulting homogenate was centrifuged at 720 g for 10 min, after which the supernatant was collected as the cytoplasmic fraction. Total protein (1 mg) was loaded in a linear sucrose gradient (5–50%) prepared with a Gradient Master former (BioComp Instruments). Samples were centrifuged for 150 min at 250,000 g at 4 °C in a SW41Ti rotor (Beckman). The gradients were analysed by Piston Gradient Fractionator (BioComp Instruments; attached to the Model EM-1 Econo UV monitor (BioRad) for continuous measurement of the absorbance at 254 nm.

**Cell proliferation assay.** Cells were plated in 96-well plates (100 µl at a concentration of 1 × 10$^4$ cells per ml). After 24 h, 0.05 mg ml$^{-1}$ MTS (3-(4,5-dimethylthiazol-2-yl)-5- (3-carboxymethoxyphenyl)-2-(4-sulphophenyl)-2H-tetrazolium, inner salt Sigma) reagent (Promega) was added to each well. The plates were incubated at 37 °C for 3 h, after which the absorbance of the solution in each well was measured at a wavelength of 492 nm.

**Crystallization and structure determination.** hCINAP-D77G mutant with His$_6$-tag was generated by PCR and the DNA coding sequence was inserted into pET-28a vector between EcoRI and XbaI restriction sites. Insertion was verified by DNA sequencing. The expression plasmid was transfected into *Escherichia coli* Rossetta cells and protein expression was induced with 0.5 mM isopropyl β- D-1-thiogalactopyranoside at 18 °C for 14 h. The recombinant protein was purified with a nickel-affinity column (Ni$^{2+}$-charged HiTrap chelating HP column from GE Healthcare), followed by a gel-filtration chromatography using a Superdex 75 column (GE Healthcare) that was previously equilibrated with buffer containing 10 mM Tris pH 8.0 and 500 mM NaCl. The fractions corresponding to hCINAP-D77G were collected and proteins were subjected to desalting with 50 mM Tris pH 7.5 and 150 mM NaCl. Next, the purified protein was concentrated to 14 mg ml$^{-1}$ for crystal screening by sitting-drop vapour diffusion method at 20 °C (Index, Natrix, PEG/Ion, Crystal screen and Crystal screen 2; Hampton Research, CA, USA). After optimization, the crystal of hCINAP-D77G was grown in solution comprising 0.01 M magnesium chloride, 0.05 M MES pH 5.6 and 1.8 M lithium sulfate. Protein crystals were stored in the mother liquor containing 20% glycerol and were flash cooled into the liquid nitrogen. All diffraction data were collected at the BL17U beamline of Shanghai Synchrotron Radiation Facility (China). Corresponding reservoir solutions with 5 and 10% (v/v) glycerol gradients were used as the cryoprotectants for data collection. Data were collected with an X-ray wavelength of 0.9792 Å and processed with HKL 2000. With the wild-type structure (PDB code: 3IIL) as a search model, both structures were solved by molecular replacement using Molrep in the CCP4 programme suite[45]. The initial models were refined using REFMAC5 (ref. 46). The final models were checked by PROCHECK.

**Co-immunoprecipitation.** To detect the role of RPS14 C-terminal tail in mediating the interaction between RPS14 and hCINAP, the four of the last eight amino acid residues (G145A, R147A, G148A and R150A) of RPS14 were mutated to alanine, respectively, or the last eight amino acid residues were deleted (1–143). 293T cells were transfected with Flag-hCINAP and the mutants or truncation of HA-RPS14. After transfection for 36 h, cells were harvested and subjected to IP assay using anti-Flag antibody. The binding of hCINAP with RPS14 was examined by western blotting with anti-HA antibody.

**Pull-down assay.** To test the effect of hCINAP on the interaction between RPS14 and 18S-E pre-rRNA, recombinant GST-RPS14 and *in vitro*-synthesized 18S-E pre-

rRNA were first added to 1 ml NP-40 lysis buffer (50 mM Tris pH 8.0, 150 mM NaCl and 1% NP-40) and incubated for 1 h at 4 °C. Next, His$_6$-hCINAP was added and incubated for another 1 h at 4 °C. Forty microlitres of glutathione S-transferase (GST) beads were then added for 1 h and finally the reaction system was incubated at room temperature for 15 min before being washed. For RNA analysis, RNA was immediately extracted and analysed on agarose gel before SYBR Green staining. To detect the effect of ATP hydrolysis on the interaction between RPS14 and hCINAP, GST-RPS14 and His-hCINAP were incubated in 1 ml NP-40 lysis buffer for 2 h at 4 °C and then ATP or AMP-PNP was added at 1 mM final concentration in the presence of MgCl$_2$ (5 mM) and incubated further for 1 h. GST beads were then added and incubated for 1 h before being washed. The interaction between RPS14 and hCINAP were examined by western blotting with the indicated antibody.

**Thin-layer chromatography.** GST-RPS14 and His-hCINAP proteins were purified using a previously described method[26]. RPS14 and hCINAP (50 ng each) were incubated with 3 µCi γ-[$^{32}$P]ATP in the buffer containing 50 mM Tris-HCl pH 7.5, 0.5 mg BSA, 5 mM DTT and 5 mM MgCl$_2$ at 37 °C for the indicated periods. The reaction was stopped by heating the mixture at 70 °C for 2 min. Samples (1 µl) were spotted on poly(ethyleneimine)-cellulose thin-layer chromatography plates (Analtech, Newwark, DE), which were developed in 0.6 M NaH$_2$PO$_4$. After drying, the plates were exposed to a phosphor-imager screen. Signals were visualized using a Typhoon PhosphorImager (GE Healthcare).

***In vitro* RNA cleavage assay.** The 18S-E pre-rRNA cleavage by Nob1was performed with the reaction buffer containing 25 mM Tris-HCl pH 7.6, 75 mM NaCl, 2 mM DTT, 100 µg ml$^{-1}$ BSA, 0.8 unit per µl RNasin, 4.5% glycerol and 5 mM MnCl$_2$. The 18S-E pre-rRNA (5′-GAUCAUUAACGGAGCCCGGGAGGGCGAGC C-3′) was 5′-radiolabelled with γ-[p$^{32}$]-ATP using the QIAquick Nucleotide Removal Kit (Qiagen, Cat.No. 28304). Before the addition of labelled 18S-E pre-rRNA, the reaction system (10 µl) containing Nob1 and hCINAP protein were pre-incubated at 37 °C for 5 min and then incubated with the 18S-E pre-rRNA for 60 min. After digestion with proteinase K for 5 min, RNA was extracted and the reaction products were resolved with denaturing 12% polyacrylamide and 8 M urea gel, and the gel was exposed to phosphor-imager screen. Signals were visualized by the Typhoon PhosphorImager.

**Cell cycle and apoptosis assay.** For the cell cycle analysis, cells were harvested by trypsinization and fixed in 75% ethanol at 4 °C overnight. Next, cells were incubated with 1 mg ml$^{-1}$ RNase A at 37 °C for 30 min, stained with propidium iodide (50 µg ml$^{-1}$) and subjected to DNA content analysis by FACS analysis. Apoptosis assays were performed using the AnnexinV-FITC Apoptosis Detection Kit (Life Technology). Cells were washed with PBS and stained with AnnexinV and propidium iodide. Apoptosis was determined by FACS analysis.

**Soft-agar colony formation and nude mice tumorigenesis.** Soft-agar colony-formation assay was performed according to the manufacturer's instructions (Cell Biolabs, Inc.). For *in vivo* tumorigenesis assays, HCT116 or U2OS cells (2 × 10$^6$ cells) were injected into the flanks of BALB/c nude mice (5–6 weeks of age, female). Tumorigenicity was monitored once per week. The mice were killed three weeks after injection of the cells. Tumours were isolated and tumour weight was measured. The nude mice tumorigenesis assay was approved by the Peking University Laboratory Animal Center. All animals were handled following the 'Guide for the Care and Use of Laboratory Animals' and the 'Principles for the Utilization and Care of Vertebrate Animals'.

**Quantitative PCR detecting the mRNA level of *hCINAP*.** Total RNA was extracted using Trizol reagent (Invitrogen). cDNA was synthesized from 1 µg of RNA with random or oligo (dT) primers using the Reverse Transcription System (Promega). Quantitative reverse transcriptase–PCR was performed with 1 µl cDNA as the template using the platinum SYBR Green Master mix (Applied Biosystems). Primer sequences of hCINAP and β-actin were as follows: hCINAP, 5′-ggTggAg TTATTgTTgATTAC-3′ and 5′-CCTTgTAggATgCTgTggC-3′; β-actin, 5′-AAgTg TgACgTggACATCCGC-3′ and 5′-CCggACTCgTCATACTCCTgCT.

**IHC and histopathological analyses.** Human breast cancer and colorectal adenocarcinoma tissue arrays were purchased from Shanghai Biochip Company Ltd (Shanghai, China). The primary antibody was rabbit anti-human hCINAP (1:4,000). Reaction products were visualized following incubation with 3, 3′-diaminobenzidine. The negative control sample was treated identically, but without the primary antibody. The percentage of hCINAP-positive cells was obtained by dividing the hCINAP-positive area by the total area and scored as follows: 0, 0–5% positive cells; 1, 6–25% positive cells; 2, 26–50% positive cells; 3, 51–75% positive cells; 4, 76–100% positive cells. The hCINAP staining intensity was measured using an imaging system consisting of a Leica DFC 420 CCD camera and a Leica DM IRE2 microscope (Leica Microsystems Imaging Solutions Ltd, Cambridge, UK). Under high magnification ( × 200), photographs of five representative fields were captured. Identical settings were applied for each photograph to remove the interference of other factors. hCINAP intensity was

scored as follows: 0, no staining; 1, weak staining; 2, mild staining; and 3, dark staining. All primary data are presented in Supplementary Data 1. Based on the combination of hCINAP staining intensity and the percentage of positive cells, all samples were classified as low hCINAP expression (product of the score of 'staining intensity' and 'percentage of positive cells' $\leq 2$) and high hCINAP expression (product of the score of 'staining intensity' and 'percentage of positive cells' $> 2$).

**Five-year overall survival analysis.** The survival analysis was performed with a colorectal adenocarcinoma tissue array purchased from Shanghai Biochip Company Ltd. All the tissue samples used in the tissue microarray were obtained from 90 patients with colorectal adenocarcinoma by way of surgery with informed consent, at Renji Hospital Affiliated to Shanghai JiaoTong University School of Medicine, from July 2006 to May 2007. For each patient, the diagnoses of colorectal adenocarcinoma were confirmed by histopathological examination and complete follow-up data were available. Representative cancer tissues without necrotic and haemorrhagic materials were collected. The tissues were washed with PBS and fixed with 4% paraformaldehyde at 4 °C overnight. Next, the samples were dehydrated and embedded in the paraffin. For each patient, one core (1 mm in diameter) were taken from the above donor paraffin blocks and transferred to the recipient blocks at defined array locations. Four-micrometre-thick sections were placed on 3-aminopropyltriethoxysilane-coated slides to construct the tissue microarray. The immunostaining of hCINAP was performed with anti-hCINAP antibody using a manufacturer recommended peroxidase anti-peroxidase method. hCINAP immunostaining was evaluated independently by three pathologists who were blinded to the outcomes of patients. The percentage of hCINAP-positive cells and staining intensity of hCINAP were evaluated as described in the above 'IHC and histopathological analyses' section. The 5-year overall survival rate of the patient with colorectal adenocarcinoma was analysed according to the expression of hCINAP in cancer tissues and the follow-up data of each patient. The survival curves were derived from Kaplan–Meier estimates and compared using log-rank tests.

**EDU (5-ethynyl-2′-deoxyuridine) staining.** HeLa cells stably expressing control-shRNA or hCINAP-shRNA were cultured in the six-well plate. Twenty-four hours later, EDU (5-Ethynyl-2'-deoxyruridine) was added to make a final concentration of 5 µM. After 48 h, cells were washed twice with PBS and fixed in 4% paraformaldehyde/PBS for 15 min at room temperature. With washing gently with PBS, cells were permeabilized with 0.2% Triton X-100 for 20 min at room temperature. Next, cells were washed with PBS and blocked with 3% BSA for 30 min. After blocking of samples, the staining mixture (70 mM Tris-HCl pH 8.0, 1 mM CuSO$_4$, 100 mM ascorbic acid (prepared fresh) and 10 µM sulfo-cyanine5-azide dye) was added to the samples and incubated for 30 min at room temperature. After washing twice with PBS, cells were counterstained with 4,6-diamidino-2-phenylindole to label the nuclei. Images were visualized with a confocal laser scanning microscope (Leica TCS SP5).

**Bioinformatic analysis of the polysome RNA-seq data.** MCF7 cells harbouring GFP–vector or GFP–hCINAP were subjected to sucrose gradient centrifugation. Ribosome profiles were obtained by measuring the absorbance of the sucrose gradient at a wavelength of 254 nm using a BioComp Gradient Fractionator. To extract RNA from the sucrose gradient fraction, 2 volumes of 8 M guanidine-HCl and 3 volumes of 99% ethanol were added. RNA was precipitated at $-20$ °C overnight and centrifuged at 38,000 g for 45 min. The supernatant was removed, after which the pellet was dried for 30 min, followed by RNA isolation with the Qiagen RNeasy Kit. The RNA samples are first treated with DNase I to degrade any possible DNA contamination. Next, the mRNA is enriched by using the oligo(dT) magnetic beads. Mixed with the fragmentation buffer, the mRNA is fragmented into short fragments. Next, the first strand of cDNA is synthesized by using random hexamer primer. Buffer, dNTPs, RNase H and DNA polymerase I are added to synthesize the second strand. The double-stranded cDNA is purified with magnetic beads. End reparation and 3′-end single nucleotide A (adenine) addition is then performed. Finally, sequencing adaptors are ligated to the fragments. The fragments are enriched by PCR amplification. The integrity and concentration of the library were evaluated by Agilent Technologies 2100 Bioanalyzer and ABI StepOnePlusReal-Time PCR System, respectively. The library products are ready for sequencing via Illumina HiSeqTM 2000. Raw reads with the sequence of adaptor, high content of unknown bases ($>10\%$) and low-quality reads were removed before data analysis. The clean reads were mapped to the UCSC hg19 reference genome and fragments per kilobase of exon per million fragments mapped was assigned to calculate expression levels. The recruitment to polysome of each mRNA was calculated according to their FPKM values in the monosome and polysome. Significantly upregulated translation with overexpressed hCINAP was defined when the recruitment to poysome has at least 1 s.d. higher from the mean difference for all the identified mRNAs. In addition, translationally inhibited mRNAs have at least 1 s.d. higher polysome recruitment in the control cells.

**Statistical analysis.** Results were showed as the mean ± s.d. from multiple independent biological replicates. P-values were obtained by Student's t-test or analysis of variance using GraphPad Prism 5.0 and SPSS 19.0 software. Survival curves were obtained from Kaplan–Meier estimates and validated with the log-rank test. P-values $< 0.05$ were considered to be statistically significant.

**Data availability.** The atomic coordinates and structural factors for hCINAP-D77G reported in this study have been deposited in the Protein Data Bank (http://www.rcsb.org/pdb) with accession code 5JZV. Sequence data have been deposited into the Gene Expression Omnibus (GEO) with the accession codes GSE81469. All the relevant data are available from the corresponding author.

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

## Acknowledgements

We thank Dr Ning Xu (Institute of Microbiology at the Chinese Academy of Sciences), Dr Jianguo Wu, Dr Renqing Feng and Dr Xin Xing (Peking University) for technical assistance. We thank Dr Lingqiang Zhang (Beijing Institute of Radiation Medicine) and Dr Lijing Wang (Guangdong Pharmaceutical University) for kindly providing MMTV-PyMT mouse. We thank Dr Yangming Wang for helpful discussions. This work was supported by grants from the National Science Foundation of China (31470754), the Doctoral Fund of the Ministry of Education of China (20130001130003) and Beijing Natural Science Foundation Grant (5152012).

## Author contributions

D.B. and X.Z. designed the experiments. D.B. performed most experiments and analysed data. J.J. provided the clinical samples. J.Z. performed the ribosome profiling assay. R.H. and X.C. performed the northern blot assay. Y.L. solved the crystal structure of hCINAP-D77G. T.L. performed the bioinformatics analysis for the RNA-seq data. Y.T. performed nude mice tumorigenesis experiment. D.H. and L.Q. helped with the statistical analysis. D.B. and X.Z. wrote the manuscript.

## Additional information

**Competing financial interests:** The authors declare no competing financial interests.

