## [Peer review file · Nature Communications]

Reviewers' comments:

Reviewer #1 (regulation of translation and cancer)

(Remarks to the Author):

The studies into CINAP are interesting and could be important. In general, the conclusions are too strong for the modest effects observed. More biochemical experiments to really demonstrate this is only about rRNA processing and not other activities really need to be presented. There is a lot of work presented, but it could be better considered and focussed to maximize its impact.

Some issues that should be addressed include

1. Why are the effects of reduction in CINAP levels not more striking when there is such an impressive effect on 18S rRNA. For instance, the effects are less than two fold on cell viability (even including the insulin experiment). Is some rescue pathway activated? Folds should be put in the text for amount of protein synthesis reduction for a given experiment.
2. CINAP acts in formation of Cajal bodies so is there an effect on splicing, as well as on regulating p53 activity, experiments should address whether these are factors in any phenotypes associated with reduction in CINAP.
3. Is there substantial motion around the mutation H79, could there be a change in dynamics rather than structure (B-factors might address this)?
4. There is no direct data demonstrating that CINAP acts in 18S rRNA processing, perhaps aberrant splicing of p53 activity indirectly led to modifications to 18S processing. Without direct biochemistry showing this, the mutational work dissecting ATPase from AKase activity is not warranted.
5. Studies with RPS14 are one possible model for how things are working, these studies would be strengthened by going through all the steps of 18S rRNA maturation and demonstrating that this specific step was affected. RPS14 is another protein with multiple functions. Further, some sort of rescue experiment with 18S rRNA would help to prove that the effects on growth are due to its effects on 18S rRNA processing.
6. Figure 4, no experiments show this is a function of the 18S rRNA processing activity of CINAP.
7. P53 is mutant in H1299 cells, again, some controls for the p53 activity of CINAP should be carried out to exclude/include this as part of the mechanisms underlying their modest phenotypes. Apoptosis only varies by 10%, it does get doubled but this is not spectacular as phenotypes go. Number of colonies are reduced by less than half, tumor volumes are more impressive but data should include tumours from both not mice in one panel and tumours in the other (4g).

8. There findings do not support conclusions such as CINAP plays a critical role in tumour development. This could be a bystander effect. Was CINAP related to disease stage? The findings related to prognosis in colon cancer are interesting, it would be interesting to know whether it was an early or late hit in the colon cancer pathway.

9. CINAP overexpression on growth is very modest. On apoptosis, the effects are between 2% and 10%, a more robust apoptotic challenge should be given and then one can observe whether CINAP can rescue cells- but when only 10% of cells die, this is a very questionable range to be in in terms of physiological relevance. All the effects of CINAP overexpression mirror the knockdown, all are very modest.

10. Does CINAP overexpression increase the 18S rRNA maturation, Cajal bodies or p53 activity?

11. Differences in 18S rRNA maturation alone are sufficient to enable ribosome specific translation? This is hard to conceptualize.

Do the authors see any specific motifs in the RNAs that could group these as USER codes?

Reviewer #2 (rRNA biogenesis, cancer and mouse work)

(Remarks to the Author):

This is a very complete paper that describes the role of the ATPase hCINAP by a multidisciplinary approach. Overall, the quality of the work is top and makes my work of reviewer very easy. I ask the authors to stick to the official nomenclature of the gene, however, at least in the abstract! Many databases do not show an entry with hCINAP.

There are a few things I would advice in order to improve the work, considered that very likely this paper will remain the "reference" for hCINAP function for a few years at least.

1. Text. There are some improvements to be made to give a clear message. For instance, the caption of the figures can be made more specific, such as Fig. 1. Rather than writing Targeted disruption of CINAP in mice, the author could directly write that it is lethal at homozygosity.

Some simplification would also help. The manuscript has many data and I understand it is difficult to focus.

Other example, Fig. 4g, obviously there is some internal discrepancy in showing on one side tumors, on the other "healthy" but dead mice. It is also rather gross for an "open access" journal. Since the point is clear ALSO without this figure I would eliminate it.

Other example, 7a, ribosome profiling is a bit misleading as it can be taken for the Rnaseq of Ingolia, whereas here it is more RNaseq on a polysomal profile. Protein translation term makes "shiver" people in the translation field which prefer the proper mRNA translation!

2. The RNaseq on ribosomes: I have missed the original raw data. Are they somewhere? I could obviously not judge the d-e panels without them. Also, are there structural elements in the mRNA affected by hCINAP at the translational level? Pertinent to this, what happens to the RNA

transcriptional profile, too? These are important statements to be made because it seems that translation is mostly affected when cells are stimulated.

3. I am not asking for this, specifically, but do the authors know if signaling to the translational machinery is affected? 4E-BP phosphorylation changes, for instance? Or the idea is that less ribosomes may affect translation? Can the author please make a strong statement on this rather important issue? See also point 5.

4. What would be nice to see is if there is a transcriptional modulation, too, and how this relates to the connectivity map as done in:

www.nature.com/ncomms/.../ncomms9261.html.

This rather simple analysis (half a day) would unveil details on the crosstalk of hCINAP with other pathways.

5. One may reason that the all effect on tumorigenesis is due to eIF4E downregulation due to Myc downregulation. These are important statements that can give a big value to scientists reading this paper in the translation or cancer field. Is eIF4E down?

6. I was a bit surprised to notice that AK6 has not been detected in the Atlas at the protein level and that it is annotated as a coiled body protein. Here (this paper) one thinks that it is perhaps nucleolar. The immunohistochemical staining does not help. Even the original panel is low resolution. My obvious questions are: where is AK6 in reality, and which are its expression levels? Are the authors 100% confident on the immunostaining they show? A wrong statement may "contaminate" databases.

7. In general. Since this paper provides a very clear idea on hCINAP function, authors should really be sure that all statements are crystal-clear and if doubts exist, better to state them (I refer to, for instance, protein localization). The work is enough good to stand lack of information if data are not perfect.

Overall, I am rather enthusiast of this piece of work.

Reviewer #3 (biochemistry of ribosome assembly)

(Remarks to the Author):

The authors carry out detailed experiments to understand how the mammalian ribosome assembly factor CINAP functions in biogenesis of small ribosomal subunits, then they explore its importance for growth of tumor cells, and why it might be important.

Mouse homozygous knockouts of CINAP are lethal, whereas heterozygotes exhibit no obvious growth defects but have a mild defect in processing of 20S pre-rRNA to mature 18S rRNA. Then the authors asked whether CINAP is more important for conditions of rapid cell growth. They observed that CINAP knockouts failed to increase growth rates upon insulin treatment. Most interestingly, CINAP seems important for growth of tumor cells. shRNA knockdowns decreased production of 18S

rRNA and 40S ribosomal subunits and polysomes as well as causing decreased protein synthesis and cell viability in tumor cells.

A detailed series of Co-IP and pull-down experiments in vivo and in vitro indicated that CINAP is an ATPase that binds to the C-terminal portion of ribosomal protein S14, and inhibits its binding to rRNA. This binding to S14 stimulates the ATPase activity of CINAP, decreasing its association with S14. CINAP can also bind to assembly factor Nob1 and stimulate its cleavage of 20S pre-rRNA to form mature 18S rRNA, which was shown to be dependent on the ATPase activity of CINAP. This led to an interesting model for how CINAP might help S14 assemble into ribosomes, coupled with processing at a nearby cleavage site in pre-rRNA. This section by itself is an interesting, timely and provocative model for coupling ribosomal protein assembly and pre-rRNA processing.

The second portion of this work demonstrates that knockdown of CINAP decreases growth of tumor cells in culture and in vivo. Consistent with a role in tumorigenesis, CINAP is expressed at higher levels in tumor cells, and its over expression promotes tumorigenesis. More specifically, this leads to selectively increased translation of mRNAs encoding cancer associated proteins.

To improve this manuscript:

(1) The authors might briefly relate their results about CINAP aiding S14 assembly to the rapidly emerging literature about chaperones dedicated to helping different ribosomal proteins assemble with rRNA, e.g., yeast rp L4, or rp L5.

(2) Could the authors provide, in Supplemental data, much more information about all of the experiments done in Figure 3?

Author response to Reviewer comments:

Reviewer #1 (regulation of translation and cancer) (Remarks to the Author):

The studies into CINAP are interesting and could be important. In general, the conclusions are too strong for the modest effects observed. More biochemical experiments to really demonstrate this is only about rRNA processing and not other activities really need to be presented. There is a lot of work presented, but it could be better considered and focussed to maximize its impact.

Some issues that should be addressed include

1. Why are the effects of reduction in CINAP levels not more striking when there is such an impressive effect on 18S rRNA. For instance, the effects are less than two fold on cell viability (even including the insulin experiment). Is some rescue pathway activated? Folds should be put in the text for amount of protein synthesis reduction for a given experiment.

Response:

18S rRNA processing and ribosome assembly are fundamental processes for cell viability. CINAP reduction inhibits 18S rRNA production and under this stressed condition, cells may initiate a redundant pathway to rescue cell growth.

Following the suggestion from the reviewer, we put the fold change in the text to clearly show the reduced amount of protein synthesis.

2. CINAP acts in formation of Cajal bodies so is there an effect on splicing, as well as on regulating p53 activity, experiments should address whether these are factors in any phenotypes associated with reduction in CINAP.

Response:

To address this comment, we conducted the following experiments to examine the effects of hCINAP on p53 splicing and activity in cells with depleted hCINAP.

- (1) The mRNA levels of the main isoforms of p53 including p53 α , β and γ are qualified by RT qPCR. The results showed that reduction in hCINAP had no significant effect on p53 splicing (Supplementary Fig.6a).
- (2) We also detected the protein abundance of different p53 isoforms using p53 antibody by western blot analysis. A slightly increased level of p53 α , but not p53 β and γ was observed with hCINAP reduction (Supplementary Fig.6b).
- (3) p53 luciferase reporter assay was performed using HCT116 cells transfected with lentivirus expressing control-shRNA or hCINAP-shRNA (MOI=8). The results showed that down-regulation of hCINAP induced an increase on p53 activity (Supplementary Fig.6c). In this study, hCINAP reduction has an impressive defect on 18S rRNA processing and ribosome assembly. It is documented that perturbations in rRNA processing are thought to cause ribosome stress and activate p53 (Zhang et al., Cancer cell, 2009, 369-377). The increases of p53 abundance and activity are probably due to the defects of ribosome assembly induced by hCINAP depletion. In addition, we also performed the following experiments to address whether the inhibition of 18S rRNA processing and cell growth by depleted hCINAP is in a p53-dependent manner.

- (1) We knocked down hCINAP in p53^{+/+} and p53^{-/-} HCT116 cells, respectively, and examined the 18S rRNA production by northern blot assay. The result showed that hCINAP depletion caused the accumulation of the precursor 18S-E pre-rRNA both in p53^{+/+} and p53^{-/-} cells, indicating that p53 deficiency has no significant effect on the reduced 18S rRNA processing induced by hCINAP-depletion (Supplementary Fig.6d).
- (2) The cell apoptosis, soft agar colony formation and nude mice tumorigenesis assays were performed with p53^{+/+} and p53^{-/-} HCT116 cells harboring control-shRNA or hCINAP-shRNA. Depletion of hCINAP in p53^{-/-} HCT116 cells induced similar cell apoptosis rate and tumorigenesis inhibition to that of p53^{+/+} HCT116 cells (Supplementary Fig.6e-g).

Taken together, the role of hCINAP in regulating 18S rRNA processing and cell growth is not dependent on p53. We added the above results as supplementary data and discussed this in the revision.

3. Is there substantial motion around the mutation H79, could there be a change in dynamics rather than structure (B-factors might address this)?

Response:

The crystal structure we showed here was the mutant hCINAP-D77G not H79G (We had tried to crystallize both mutants, but only have obtained the crystal and solved the dimensional structure of hCINAP-D77G as shown in Fig. 3b). The structural superposition of hCINAP-D77G (PDB code: 4QSO) with hCINAP-ADP (PDB code: 3IIL) showed no substantial motion around the mutation D77G (Fig.3b). The relative B factor value of amino acid at position 77 over R70-D82 region is 0.88 for G77 in the hCINAP-D77G mutant structure and 0.90 for D77 in the hCINAP-ADP wild-type structure. Therefore, there is no obvious B-factor change brought by the mutation D77G.

4. There is no direct data demonstrating that CINAP acts in 18S rRNA processing, perhaps aberrant splicing of p53 activity indirectly led to modifications to 18S processing. Without direct biochemistry showing this, the mutational work dissecting ATPase from AKase activity is not warranted.

Response:

As our response to Question 2, to detect whether the role of hCINAP in 18S rRNA processing indirectly caused by p53, northern blot assay was performed to examine the effect of depleted hCINAP on 18S rRNA processing in p53^{-/-} HCT116 cells. The results showed that depletion of hCINAP led to the accumulation of 18S-E pre-rRNA in the absence (or presence) of p53 (Supplementary Fig.6d). This data suggested that regulations to 18S rRNA processing by hCINAP reduction is not through regulating p53.

Moreover, to demonstrate the direct involvement of hCINAP in 18S rRNA processing, we reviewed related literatures that defined processing factors directly involved in 18S rRNA processing (Simon et al., NSMB, 2012; Emmanuel et al., EMBO J, 2008; Sander et al., MCB, 2005) and performed the following experiments.

- (1) We examined the direct interaction between hCINAP and the 18S rRNA precursor 18S-E pre-rRNA. RNA-EMSA was performed with purified hCINAP and *in vitro* synthesized 18S-E pre-rRNA. The results showed that hCINAP specifically binds the 18S-E rRNA but not the matured 18S rRNA.

And the enzymatic mutation of hCINAP D77G or H79G disrupted the interaction between hCINAP with 18S-E pre-rRNA (Supplementary Fig.3f). This result indicated the hCINAP directly binds to 18S-E pre-rRNA.

- (2) Consistent with the *in vitro* EMSA data, IP (Immunoprecipitation)-RT-PCR assays were also performed with 293T cells expressing Flag-vector or Flag-hCINAP. Cells were harvested and IP was carried out using anti-Flag antibody. The binding of hCINAP to 18S-E pre-rRNA was examined by PCR. The results showed that hCINAP binds to the 18S-E pre-rRNA *in vivo* (Supplementary Fig.3e).
- (3) We also performed northern blot assay and found that depletion of hCINAP caused the accumulation of 18S-E pre-rRNA (Fig. 2h) accompanied by the decreased level of 18S rRNA (Fig. 2i).
- (4) Beyond the above data, the more convincing data revealing the direct role of hCINAP in 18S rRNA processing is the *in vitro* rRNA cleavage assay shown in Fig.3h and Supplementary Fig.3i. This experiment was performed with purified hCINAP and synthesized 18S-E pre-rRNA. The cell-free system demonstrated that hCINAP directly involved in Nob1-mediated 18S rRNA cleavage. We also examined the purified hCINAP and Nob1 proteins by SDS-PAGE.

Collectively, combining the above data and related literature, our results demonstrated that hCINAP directly functions in 18S rRNA processing.

5. Studies with RPS14 are one possible model for how things are working, these studies would be strengthened by going through all the steps of 18S rRNA maturation and demonstrating that this specific step was affected. RPS14 is another protein with multiple functions. Further, some sort of rescue experiment with 18S rRNA would help to prove that the effects on growth are due to its effects on 18S rRNA processing.

Response:

Thanks for the helpful suggestion. We performed northern blot assay with the 5'-ITS1 probe that targets all the precursors of 18S rRNA to go through which processing step was specifically affected by hCINAP reduction. The result showed that depletion of hCINAP caused significant accumulation of 18S-E rRNA (Fig.2h). Notably, our result is much similar to the increased 18S-E rRNA caused by depletion of the endonuclease Nob1 (Milena et al., NAR, 2013). Nob1 specifically functions in catalyzing the final cleavage of 18S-E rRNA into mature 18S rRNA. From the results of Milena *et al* and ours, we noticed that an increased 26S rRNA was also accumulated with Nob1 or hCINAP reduction, suggesting that 18S-E rRNA processing defects may be as a negative feedback leading to the accumulation of the precursors of 18S-E rRNA. The accumulation of 18S-E pre-rRNA by hCINAP reduction supported the conclusion that hCINAP functions in promoting 18S-E rRNA cleavage into mature 18S rRNA.

To prove the growth defects induced by depleted hCINAP are due to its effects on 18S rRNA processing, the matured 18S rRNA was extracted from cells and then transfected into hCINAP-depleted cells. As the transfected 18S rRNA may be degraded quickly, we performed EDU (5-ethynyl-2'-deoxyuridine) staining to examine whether 18S rRNA transfection could rescue the defects in cell proliferation induced by hCINAP reduction. The result showed that hCINAP depletion dramatically decreased the cell proliferation rate and rescue of 18S rRNA production significantly recovered the cell proliferation defects (Supplementary Fig. 4a,b).

Besides, the observation that depletion of hCINAP caused defects in 18S rRNA processing is very similar to that of Nob1 reduction. Several publications have shown that knockdown of Nob1

expression significantly promotes apoptosis and inhibits cell proliferation (Yin, et al., *Oncol Rep.* 2015. 34:3077-3087; Li et al., *Oncol Rep.* 2014. 31:1271-1276; Wang, et al., *Gene.* 2013.528:146-153). Therefore, to further demonstrate the growth defects by hCINAP depletion was mainly due to its function in 18S rRNA processing, we knocked down hCINAP and Nob1, respectively, and compared the cell growth inhibition. The results showed that depletion of hCINAP and Nob1 caused similar cell growth defects (Supplementary Fig.4c). As the dominant role of Nob1 is catalyzing 18S-E pre-rRNA into mature 18S rRNA and based on the similar role of hCINAP in 18S rRNA processing, the growth defects by hCINAP reduction is mostly likely through affecting 18S rRNA maturation.

Taken together, these results suggested that the cell growth inhibition by hCINAP reduction is mainly due to its function in 18S rRNA processing.

6. Figure 4, no experiments show this is a function of the 18S rRNA processing activity of CINAP.

Response:

In this study, depletion of hCINAP disrupts 18S rRNA processing, ribosome assembly and finally inhibits total protein synthesis up to 40%. For the rapid cancer cell growth, such severe defects in protein synthesis could lead to cell growth inhibition.

Following the suggestion from reviewer, to evaluate whether the growth defects caused by hCINAP depletion was associated with p53, we performed analyses of cell apoptosis, soft agar colony formation as well as nude mice tumorigenesis using p53^{+/+} and p53^{-/-} HCT116 cells with depleted hCINAP. The results showed that similar cell growth inhibition was observed with hCINAP depletion in the absence or presence of p53 (Supplementary Fig.6e-g), indicating that the effect of hCINAP on cell growth is not through regulating p53.

And following the reviewer's suggestion, we also rescued the 18S rRNA production in cells with depleted hCINAP and find that rescue of 18S rRNA obviously recovered the cell growth inhibition by hCINAP reduction (Supplementary Fig.4a,b). Meanwhile, we showed that hCINAP and Nob1 reduction caused a similar cell growth inhibition (Supplementary Fig.4c). Therefore, the cell growth defects by hCINAP depletion mainly results from its dysfunction in 18S rRNA processing.

As hCINAP may have multiple roles beyond regulating 18S rRNA processing, we also agreed that other unknown physiological functions of hCINAP might also contribute to the growth defects induced by hCINAP depletion. We discussed this in the related context and added these results as supplementary data in the revision.

7. P53 is mutant in H1299 cells, again, some controls for the p53 activity of CINAP should be carried out to exclude/include this as part of the mechanisms underlying their modest phenotypes. Apoptosis only varies by 10%, it does get doubled but this is not spectacular as phenotypes go. Number of colonies are reduced by less than half, tumor volumes are more impressive but data should include tumours from both not mice in one panel and tumors in the other (4g).

Response:

To address this comment, we evaluated the involvement of p53 in the phenotypes caused by hCINAP reduction. As mentioned in Questions 2 and 6, hCINAP reduction in p53^{-/-} HCT116 cells promotes cell apoptosis, inhibits soft agar colony formation and nude mice tumorigenesis, which has no significant difference with that of depletion of hCINAP in p53^{+/+} HCT116 cells. These results demonstrated that cell growth defect by hCINAP depletion is not dependent on p53.

As the reviewer concerned, compared with the modest phenotype in cell apoptosis by hCINAP reduction, the cell growth defects in nude mice tumorigenesis is more impressive. We think it is likely due to the difference of the culture time for the hCINAP-depleted cells. The apoptosis assay was performed at 48 h after cells transfected with lentivirus expressing hCINAP-shRNA. hCINAP reduction inhibits ribosome assembly, however, as basal ribosome amount is large, the inhibition of ribosome assembly by hCINAP depletion is not so severely to cells at the early stage, so dramatic cell apoptosis rate was hard to be observed. While for the nude mice tumorigenesis assay, cells were seeded in the mice and cultured for three weeks. During the relative longer period, defects of ribosome assembly were accumulated and the inhibition to cell growth was amplified.

To prove the above idea, HCT116 cells expressing hCINAP-shRNA were cultured for a relative longer time (one week) and the cell apoptosis was assessed. The result showed that compared with control cells, depletion of hCINAP increased the cell apoptosis rate by 20% (Fig.4c, d). *In vitro* soft agar colony formation assay was also carried out using HCT116 cells with depleted hCINAP, and the data was shown in Fig.4e in the revision.

For the nude mice tumorigenesis assay shown in Fig.4g (previous version), to show the results in a complete image, we repeated this assay with HCT116 cells harboring control-shRNA or hCINAP-shRNA. The results also showed that depletion of hCINAP abolished tumorigenesis and no tumors were found in nude mice injected with hCINAP-depleted HCT116 cells. The tumors formed by control cells were isolated as well. We updated this data as Fig.4f in the revision.

8. There findings do not support conclusions such as CINAP plays a critical role in tumour development. This could be a bystander effect. Was CINAP related to disease stage? The findings related to prognosis in colon cancer are interesting, it would be interesting to know whether it was an early or late hit in the colon cancer pathway.

Response:

Following the reviewer's suggestion, we deleted the statement that "highly expressed hCINAP in cancer promotes tumor development" in the manuscript. Also, to explore the role of hCINAP in tumorigenesis, the CINAP^{+/+} and CINAP^{+/-} mice were crossed with MMTV-polyomavirus middle T antigen (PyMT) transgenic mice, which have been widely used to study mammary tumorigenesis (Actually this experiment started 4 months ago, which took a long time to get the result). The results showed that all the PyMT/CINAP WT spontaneously developed breast tumors at 70 days of age. However, the PyMT/CINAP^{+/-} mice started developing tumors at 90 days and only 40% (2/5) of the mice developed tumors at 120 days (Supplementary Fig.4d). These data suggest that hCINAP plays important roles in regulating tumor growth.

Our data in Fig. 5c assessed the expression levels of hCINAP in different colon cancer stages from 74 patients by immunochemical staining (IHC). To address the concern from the reviewer, we re-analyzed the expression levels of hCINAP in I, II and III stages of colon cancer tissues in details. A slightly increased hCINAP expression was observed with the development of colon cancer (Supplementary Fig.5g). Since only 74 samples were available in this study, a substantial increase in the number of samples is needed to accurately evaluate whether hCINAP expression is significantly correlated with the disease development in the future study.

9. CINAP overexpression on growth is very modest. On apoptosis, the effects are between 2% and 10%, a more robust apoptotic challenge should be given and then one can observe whether CINAP can rescue cells- but when only 10% of cells die, this is a very questionable range to be in in terms of physiological relevance. All the effects of CINAP overexpression mirror the knockdown, all are very modest.

Response:

Thanks for the reviewer's helpful suggestion. To address this concern, the chemotherapy drug cisplatin (10 µg/ml) was used to jump the cell apoptosis rate, and the effect of hCINAP overexpression on cisplatin-induced cell apoptosis was evaluated. The results showed that hCINAP overexpression resisted to cisplatin-induced cell apoptosis (24%, Fig. 6c,d).

For the modest effect of hCINAP overexpression on cellular growth, we think one of the reasons is that the endogenous level of hCINAP has been elevated in cancer cells, thus overexpression of hCINAP may not work such efficiently compared with that of knockdown. To avoid the dominant influence of endogenous hCINAP, we selected U2OS cells in which endogenous level of hCINAP is relative lower than that of MCF7 and HCT116 cells. U2OS cells were transfected with lentivirus expressing Flag-vector and Flag-hCINAP and the following experiments were performed:

- (1) MTT assay were performed with U2OS cells expressing Flag-vector and Flag-hCINAP. The results showed that highly expressed hCINAP promotes cell growth (updated as Fig.6a).
- (2) Cell cycle progression was analyzed by FACS. The result showed that highly expressed hCINAP significantly promotes the G1 cell cycle progression (updated as Fig.6b).
- (3) *In vitro* colony formation assay was performed with U2OS cells stably expressing Flag-vector and Flag-hCINAP. The result showed that U2OS cells with overexpression of hCINAP generated more and larger clones compared with that control cells (updated as Fig.6e).
- (4) Nude mice tumorigenesis assay was repeated with U2OS cells and overexpression of hCINAP significantly promotes tumor growth. The tumor weight of cells expressing Flag-hCINAP is about 3-fold to that of control cells (updated as Fig.6f,g).

Taken together, overexpression of hCINAP in U2OS cells promotes cancer cell growth. And the extent of promotion effect of hCINAP on cell growth is similar to other observations in published literature (Lu et al., Cancer cell, 2011).

10. Does CINAP overexpression increase the 18S rRNA maturation, Cajal bodies or p53 activity?

Response:

To address these comments, northern blot assay was performed using 18S rRNA probe to detect the effect of overexpressed hCINAP on 18S rRNA maturation. Increased 18S rRNA maturation was observed in cells with overexpression of hCINAP (Supplementary Fig.8f).

We also performed immunofluorescence assay with HeLa cell to detect if hCINAP overexpression significantly affects Cajal bodies formation. The number of Cajal body in the nucleus was counted from 100 nuclei resulting from more than 10 images from cell transfecting with Flag-vector or Flag-hCINAP, respectively. The results showed that no obvious change was observed in the number of Cajal bodies with hCINAP overexpression compared with that of control cells (Supplementary Fig.8a,b).

To detect the p53 abundance in cells with overexpressed hCINAP, HCT116 cells were transfected with pCMV-p53 and an increased amount of Flag-hCINAP. We also performed luciferase assay to measure the p53 activity with overexpression of hCINAP. The result showed that overexpression of hCINAP has no significant effect on the abundance and activity of p53 (Supplementary Fig. 8c,d).

11. Differences in 18S rRNA maturation alone are sufficient to enable ribosome specific translation? This is hard to conceptualize.

Do the authors see any specific motifs in the RNAs that could group these as USER codes?

Response:

Defects in 18S rRNA processing directly affect ribosome assembly and then influence translation pattern. It has been reported that the affinity of ribosome for any single mRNA species is unique (Lodish et al., 1974, Nature, 385-388). Given that there is an excess in the number of mRNA transcripts to ribosome, a decrease in ribosome number would affect not only on the rate of translation, but also on the patterns of translation (Thomas, 2000, Nat Cell Bio, E71-72). When defects in 18S rRNA maturation occur, ribosome assembly is decreased, and those mRNAs have higher affinity to ribosome will continue to be translated, whereas the translation of those mRNAs have lower affinity to ribosome will decrease (Ruggero et al., 2003, Nat Rev Cancer, 179-192). Changes in specific gene expression caused by alterations in ribosome number have been implicated in aberrant growth and human pathologies (Zhang et al., 2009, Cancer Cell, 369-377).

To identify the specific motifs in the RNAs which were translationally activated by hCINAP, MEME motif elicitation software was used to seek the enriched motifs (up to 12 nt for motif lengths) in the 5' UTR region of those RNAs. The result showed that the most significantly enriched motif is G/C-rich motif in the form of C(GGC)₃GG, which is highly suggestive of G-quadruplex formation. Interestingly, our result is consistent with recent studies of translation regulation brought by eIF4A1 or eIF6 reduction (Brina et al., Nature Communications, 2015; Modelska et al., Cell Death and Disease, 2015). Depletion of eIF4A1 or eIF6 in cancer cells led to translation downregulation of mRNAs enriched for G-quadruplex-forming sequences. In our study, the specific GC-rich motif in the mRNAs with upregulated translational rate suggests that highly expressed hCINAP may preferentially promote the translation of these mRNAs. We added this data as Fig. 7g in the revision.

Reviewer #2 (rRNA biogenesis, cancer and mouse work)(Remarks to the Author):

This is a very complete paper that describes the role of the ATPase hCINAP by a multidisciplinary approach. Overall, the quality of the work is top and makes my work of reviewer very easy. I ask the authors to stick to the official nomenclature of the gene, however, at least in the abstract! Many databases do not show an entry with hCINAP.

There are a few things I would advise in order to improve the work, considered that very likely this paper will remain the "reference" for hCINAP function for a few years at least.

1. Text. There are some improvements to be made to give a clear message. For instance, the caption of the figures can be made more specific, such as Fig. 1. Rather than writing Targeted disruption of CINAP in mice, the author could directly write that it is lethal at homozygosity.

Some simplification would also help. The manuscript has many data and I understand it is difficult to focus.

Other example, Fig. 4g, obviously there is some internal discrepancy in showing on one side tumors, on the other "healthy" but dead mice. It is also rather gross for an "open access" journal. Since the point is clear ALSO without this figure I would eliminate it.

Other example, 7a, ribosome profiling is a bit misleading as it can be taken for the Rnaseq of Ingolia, whereas here it is more RNAseq on a polysomal profile. Protein translation term makes "shiver" people in the translation field which prefer the proper mRNA translation!

Response:

Thanks for the helpful suggestions. As suggested by the reviewer, we added the official nomenclature of hCINAP as "Adenylate kinase (AK6)" in the abstract.

We changed the statement of Fig.1 as "Disruption of CINAP results in embryonic lethality".

For Fig.4g (previous version), to show the results in a complete image, we repeated this assay with HCT116 cells harboring control-shRNA or hCINAP-shRNA. The results also showed that depletion of hCINAP abolished tumorigenesis and no tumors were found in nude mice injected with hCINAP-depleted HCT116 cells. The tumors formed by control cells were isolated as well. We updated this data as Fig.4f in the revision.

The statement for Fig.7a was also corrected as "RNAseq on polysomal profile". Besides, we also changed "Protein translation" to "mRNA translation" in the revision.

In addition, as the reviewer suggested, we made a strong statement in the revision that in our case, the changes in ribosome assembly induced by hCINAP affect translation.

2. The RNAseq on ribosomes: I have missed the original raw data. Are they somewhere? I could obviously not judge the d-e panels without them. Also, are there structural elements in the mRNA affected by hCINAP at the translational level? Pertinent to this, what happens to the RNA transcriptional profile, too? These are important statements to be made because it seems that translation is mostly affected when cells are stimulated.

Response:

We added the original data for RNAseq on polysomal profile as Supplementary Table 4 in the revision.

To seek the specific motifs in the RNAs that were translationally regulated by hCINAP, MEME motif elicitation software was used to identify the enriched motifs (up to 12 nt for motif lengths) in the 5'UTR region of those RNAs. The most significantly enriched motif is G/C-rich motif in the form

of C(GGC)₃GG, which is highly suggestive of G-quadruplex formation. Our results is consistent with a recent study of translation regulation brought by eIF6 reduction, and polysomes from eIF6 depleted cells showed reduction of G/C rich of mRNA (Brina et al., Nature Communications, 2015). It has been reported that CG rich mRNA are generally translated with poor efficiency and growth factors such as insulin stimulate the translation of GC containing mRNA that encode for proteins necessary for cell cycle progression (Mamane et al., 2006, Oncogene, 6416-6422). In our study, the specific GC-rich motif in the mRNAs with upregulated translational rate suggests that highly expressed hCINAP may preferentially promote the translation of these mRNAs.

For the RNA transcriptional profile, we also performed RNAseq analysis with cells expressing GFP-vector or GFP-hCINAP. GO analysis showed that highly expressed hCINAP induced a gene-expression signature marked by upregulating cation channel complex, sexual production and neurotransmitter metabolic process (Supplementary Fig.7; Supplementary Table 6). Future study may be desired to explore the functions of hCINAP in these physiological processes and identify the coupling relationship of translation and transcription regulated by hCINAP.

3. I am not asking for this, specifically, but do the authors know if signaling to the translational machinery is affected? 4E-BP phosphorylation changes, for instance? Or the idea is that less ribosomes may affect translation? Can the author please make a strong statement on this rather important issue? See also point 5.

Response:

Following the reviewer's suggestion, we detected the effect of hCINAP on the level of 4E-BP1 phosphorylation in cells with overexpressed or depleted hCINAP. The results showed that no significant difference was observed for the phosphorylation level of 4E-BP1 in either case (Supplementary Fig. 6h,8e). As suggested by the reviewer, we made a strong statement in the revision that the effect of hCINAP on the translation is mainly due to the regulation of ribosome assembly by hCINAP.

4. What would be nice to see is if there is a transcriptional modulation, too, and how this relates to the connectivity map as done in:

www.nature.com/ncomms/.../ncomms9261.html.

This rather simple analysis (half a day) would unveil details on the crosstalk of hCINAP with other pathways.

Response:

Thanks for the reviewer's nice suggestion. We performed Connectivity Map analysis with the mRNAs with significantly changed translation rate to discover the connections between drugs and gene expression. Highly expressed hCINAP promotes cancer cell growth, and consistent with this, hCINAP overexpression "signature" showed a strong negative correlated with the results obtained by two anti-cancer drugs, verteporfin and lycorine (Supplementary Table 5). Verteporfin was exhibited to inhibit cancer progression partially by impairing the global clearance of high- molecular weight of oligomerized proteins including p62 and STAT3 (Zhang et al., Sci. Signal. 2015; Perra et al., J Hepatol. 2014, 1088-1096; Feng et al., Cancer cell, 2014,831-845). Lycorine has the inhibitory effects on

proliferation and invasion of leukemia cells (Nair and van Staden, *Natural product communications*, 3014,1193-1210). These results suggested a possible crosstalk of hCINAP with the downstream signaling pathways of verteporfin and lycorine.

5. One may reason that the all effect on tumorigenesis is due to eIF4E downregulation due to Myc downregulation. These are important statements that can give a big value to scientists reading this paper in the translation or cancer field. Is eIF4E down?

Response:

To address this concern from the reviewer, we detected the protein abundance of eIF4E in cells expressing control-shRNA or hCINAP-shRNA. Compared with the protein level of eIF4E in cells with control-shRNA, no significant decreased of eIF4E was observed with hCINAP reduction. This experiment was performed three times and similar data were obtained. We added the data as Supplemental Fig. 6h and discussed this in the revision.

6. I was a bit surprised to notice that AK6 has not been detected in the Atlas at the protein level and that it is annotated as a coiled body protein. Here (this paper) one thinks that it is perhaps nucleolar. The immunohistochemical staining does not help. Even the original panel is low resolution. My obvious questions are: where is AK6 in reality, and which are its expression levels? Are the authors 100% confident on the immunostaining they show? A wrong statement may "contaminate" databases.

Response:

To answer the reviewer's question, we performed immunofluorescence staining and subcellular fractionation assay to detect the intercellular localization of endogenous hCINAP. The results showed that hCINAP is observed both in the cytoplasm and nucleus (Supplementary Fig.5d,e). We also quantified the expression level of hCINAP in the nucleus and cytoplasm with the cancer tissue assay (Supplementary Table 3). Significant difference was observed for the expression level of hCINAP only in the cytoplasm but not the nucleus between cancer tissue and adjacent cancer tissues (Supplementary Fig. 5f). We make a clear statement that hCINAP localized both in the cytoplasm and nucleus and the cytoplasmic hCINAP is highly expressed in human cancers. The original data for IHC assay was reorganized as Supplementary Table 3 in the revision.

7. In general. Since this paper provides a very clear idea on hCINAP function, authors should really be sure that all statements are crystal-clear and if doubts exist, better to state them (I refer to, for instance, protein localization). The work is enough good to stand lack of information if data are not perfect.

Overall, I am rather enthusiast of this piece of work.

Reviewer #3 (biochemistry of ribosome assembly)(Remarks to the Author):

The authors carry out detailed experiments to understand how the mammalian ribosome assembly factor CINAP functions in biogenesis of small ribosomal subunits, then they explore its importance for growth of tumor cells, and why it might be important.

Mouse homozygous knockouts of CINAP are lethal, whereas heterozygotes exhibit no obvious growth defects but have a mild defect in processing of 20S pre-rRNA to mature 18S rRNA. Then the authors asked whether CINAP is more important for conditions of rapid cell growth. They observed that CINAP knockouts failed to increase growth rates upon insulin treatment. Most interestingly, CINAP seems important for growth of tumor cells. shRNA knockdowns decreased production of 18S rRNA and 40S ribosomal subunits and polysomes as well as causing decreased protein synthesis and cell viability in tumor cells.

A detailed series of Co-IP and pull-down experiments in vivo and in vitro indicated that CINAP is an ATPase that binds to the C-terminal portion of ribosomal protein S14, and inhibits its binding to rRNA. This binding to S14 stimulates the ATPase activity of CINAP, decreasing its association with S14. CINAP can also bind to assembly factor Nob1 and stimulate its cleavage of 20S pre-rRNA to form mature 18S rRNA, which was shown to be dependent on the ATPase activity of CINAP. This led to an interesting model for how CINAP might help S14 assemble into ribosomes, coupled with processing at a nearby cleavage site in pre-rRNA. This section by itself is an interesting, timely and provocative model for coupling ribosomal protein assembly and pre-rRNA processing.

The second portion of this work demonstrates that knockdown of CINAP decreases growth of tumor cells in culture and in vivo. Consistent with a role in tumorigenesis, CINAP is expressed at higher levels in tumor cells, and its over expression promotes tumorigenesis. More specifically, this leads to selectively increased translation of mRNAs encoding cancer associated proteins.

To improve this manuscript:

(1) The authors might briefly relate their results about CINAP aiding S14 assembly to the rapidly emerging literature about chaperones dedicated to helping different ribosomal proteins assemble with rRNA, e.g., yeast rp L4, or rp L5.

Response:

As the reviewer suggested, we referred the related paper (Pausch et al., Nature Communications 6, 7494 (2015); Pillet, B., et al., PLoS Genetics 11, e1005565 (2015); Kressler, D., et al. Science 2012, 666-671). From these studies, the authors revealed that ribosomal proteins are captured by the dedicated chaperones (in a co-translational pattern) or transport system to assure their stable expression, correct subcellular localization and finally correct assembly into the pre-ribosomal particles. In our study, hCINAP chaperones RPS14 assembly into pre-40S particles through transient binding with RPS14 to inhibit premature interaction between RPS14 and 18S pre-rRNA, which is assumed to facilitate the conformational change of 18S pre-rRNA that allows proper incorporation of RPS14. We related our result with the studies suggested by reviewer in the Discussion section in the revision.

(2) Could the authors provide, in Supplemental data, much more information about all of the experiments done in Figure 3?

Response:

Following the reviewer's suggestion, we provided detailed experimental methods about all the experiments done in Fig.3. in the Supplemental Information.

REVIEWERS' COMMENTS:

Reviewer #1 (Remarks to the Author):

The authors have made substantial efforts to address my concerns. I note that counting Cajal bodies is not any measure of their activity and thus the authors should state in the text that they cannot rule out that some of their biological effects are due to splicing alterations. The 18S rRNA rescue and p53 experiments reduce my concerns that the 18S effects were not key in the underlying effects observed.

All in all, my comments have been addressed well.

Reviewer #2 (Remarks to the Author):

This manuscript has improved after the resubmission and I find it good.

A few things that I noticed in the revised version.

Figure 7h, authors in the Figure body still show protein translation. As previously stated, it is more appropriate mRNA translation.

Figure 5h and data: the legend is wrong. Authors label 5h in the body, but they write a legend for 5f. What is "competed" in the Kaplan-Meier? Importantly, I have not found an explanation of which kind of tumors were (beside colorectal), how they were collected and analyzed. The authors should describe better the patient set, perhaps in a specific method paragraph, because I missed this information.

Figure 4f, as I think I stated before, I am not sure that it is appropriate to show mice here. Also, please be sure to refer to ethical guidelines employed in the experiment for sacrifice and so on.

Supplementary Figure 5d. Scale bar definition is absent.

Author response to Reviewer comments:

Reviewer #1 (Remarks to the Author):

The authors have made substantial efforts to address my concerns. I note that counting Cajal bodies is not any measure of their activity and thus the authors should state in the text that they cannot rule out that some of their biological effects are due to splicing alterations. The 18S rRNA rescue and p53 experiments reduce my concerns that the 18S effects were not key in the underlying effects observed.

All in all, my comments have been addressed well.

Reviewer #2 (Remarks to the Author):

This manuscript has improved after the resubmission and I find it good.

A few things that I noticed in the revised version.

Figure 7h, authors in the Figure body still show protein translation. As previously stated, it is more appropriate mRNA translation.

Response:

Thanks for pointing out the mistake. 'protein translation' in Fig.7h has been changed to 'mRNA translation'.

Figure 5h and data: the legend is wrong. Authors label 5h in the body, but they write a legend for 5f. What is "competed" in the Kaplan-Meyer? Importantly, I have not found an explanation of which kind of tumors were (beside colorectal), how they were collected and analyzed. The authors should describe better the patient set, perhaps in a specific method paragraph, because I missed this information.

Response:

Thank you very much for pointing out the error. The legend has been corrected as Fig.5h in the manuscript.

The data shown in Fig.5h demonstrated that patients with colorectal adenocarcinoma bearing tumors with high level of hCINAP showed poor overall survival (OS) than those bearing tumors with low level of hCINAP.

The survival analysis was performed with a colorectal adenocarcinoma tissue array purchased from Shanghai Biochip Company Ltd. (Shanghai, China). All the tissue samples used in the tissue microarray (TMA) obtained from 90 patients with colorectal adenocarcinoma by way of surgery with informed consent at Renji Hospital Affiliated to Shanghai JiaoTong University School of Medicine from July 2006 to May 2007. For each patient, complete follow-up data were available and the diagnoses of colorectal adenocarcinoma were confirmed by histopathological examination. Representative cancer tissues without necrotic and hemorrhagic materials were collected. The tissue were washed with PBS and fixed with 4% paraformaldehyde at 4°C overnight. And then the samples were dehydrated, embedded in the paraffin. For each patient, one core (1 mm in diameter) were taken from the above donor paraffin blocks and transferred to the recipient blocks at the defined

array positions. Sections of 4 μm thickness were placed on 3-aminopropyltriethoxysilane-coated slides to construct tissue microarray. The immunostaining of hCINAP was performed with anti-hCINAP antibody using a manufacture recommended peroxidase anti-peroxidase (PAP) method. hCINAP immunoreactivity was evaluated independently by 3 pathologists who were blinded to the patient outcomes. The percentage of hCINAP-positive cells was obtained by dividing the hCINAP-positive area by the total area and scored as follows: 0, 0–5% positive cells; 1, 6–25% positive cells; 2, 26–50% positive cells; 3, 51–75% positive cells; 4, 76–100% positive cells. The hCINAP staining intensity was measured using an imaging system consisting of a Leica DFC 420 CCD camera and a Leica DM IRE2 microscope (Leica Microsystems Imaging Solutions Ltd., Cambridge, United Kingdom). Under high magnification ($\times 200$), photographs of five representative fields were captured. Identical settings were applied for each photograph to remove the interference of other factors. hCINAP intensity was scored as follows: 0, no staining; 1, weak staining; 2, mild staining; and 3, dark staining. All primary data are presented in Supplementary Data 1. Base on the combination of hCINAP staining intensity and positive cells, all samples were classified as low hCINAP expression (product of the score of 'staining intensity' and 'percentage of positive cells' ≤ 2) and high hCINAP expression (product of the score of 'staining intensity' and 'percentage of positive cells' > 2). The five-year overall survival rate of the patient was analyzed according to the expression of hCINAP in cancer tissues and the follow-up data of each patient. The survival curves were derived from Kaplan-Meier estimates and compared using log-rank tests. We added this as a separated paragraph in the section of Methods.

Figure 4f, as I think I stated before, I am not sure that it is appropriate to show mice here. Also, please be sure to refer to ethical guidelines employed in the experiment for sacrifice and so on.

Response:

We think carefully about the reviewer's suggestions. In our case, depletion of hCINAP abolished tumor growth and only cells harboring control-shRNA generated tumors. If mice were not shown, and only tumors of control cells can be presented. Therefore, mice shown in Fig.4f could make the result more completed and convincing. We also reviewed related literature and found some of them also showed mice attached to the isolated tumors. For another concern, we made a statement in the figure legend and methods about the ethical guidelines related to this experiment.

Supplementary Figure 5d. Scale bar definition is absent.

Response:

As the reviewer suggested, the scale bar related to Supplementary Fig. 5d has been added in the legend.